# Neurons throughout the brain embed robust signatures of their anatomical location into spike trains

**Gemechu Bekele Tolossa[1†], Aidan M Schneider[1†], Eva Dyer[2], Keith B Hengen[1]***

[1]Department of Biology, Washington University in St Louis, St Louis, United States;
[2]Department of Biomedical Engineering, Georgia Institute of Technology, Atlanta, United States

## eLife Assessment

This paper provides a **useful** systematic quantification of the relationship between electrophysiological response properties of single neurons with their position in the brain. The quality of the classification setup is high and the methodology is **solid**.

***For correspondence:**
khengen@wustl.edu

[†] These authors contributed equally to this work.

**Competing interest:** The authors declare that no competing interests exist.

**Abstract** Neurons in the brain are known to encode diverse information through their spiking activity, primarily reflecting external stimuli and internal states. However, whether individual neurons also embed information about their own anatomical location within their spike patterns remains largely unexplored. Here, we show that machine learning models can predict a neuron's anatomical location across multiple brain regions and structures based solely on its spiking activity. Analyzing high-density recordings from thousands of neurons in awake, behaving mice, we demonstrate that anatomical location can be reliably decoded from neuronal activity across various stimulus conditions, including drifting gratings, naturalistic movies, and spontaneous activity. Crucially, anatomical signatures generalize across animals and even across different research laboratories, suggesting a fundamental principle of neural organization. Examination of trained classifiers reveals that anatomical information is enriched in specific interspike intervals as well as responses to stimuli. Within the visual isocortex, anatomical embedding is robust at the level of layers and primary versus secondary but does not robustly separate individual secondary structures. In contrast, structures within the hippocampus and thalamus are robustly separable based on their spike patterns. Our findings reveal a generalizable dimension of the neural code, where anatomical information is multiplexed with the encoding of external stimuli and internal states. This discovery provides new insights into the relationship between brain structure and function, with broad implications for neurodevelopment, multimodal integration, and the interpretation of large-scale neuronal recordings. Computational approximations of anatomy have the potential to support in vivo electrode localization.

## Introduction

Foundational to any effort towards understanding the brain is a singular question: what information is carried in a neuron's spiking? It is widely understood that the action potential is the unit of information exchange in the central nervous system. Adrian first recorded a single neuron's action potential in 1928, establishing a rate code in sensory neurons (*Adrian and Bronk, 1928*). In 1943, McCulloch and Pitts demonstrated that neuronal circuits could compute Boolean algebra (*McCulloch and Pitts, 1943*), and 16 years later showed that the visual environment is conveyed to the brain by way of neuronal spike patterns (*Lettvin et al., 1959*). Since then, the concept of a neural code has

emerged—neuronal spiking is determined by inputs, including stimuli, and noise (*Faisal et al., 2008*; *Gerstner et al., 2014*; *Mainen and Sejnowski, 1995*; *Softky and Koch, 1993*). In parallel works spanning 1899–1951, Cajal, Brodmann, and Penfield demonstrated diverse neuronal types that organize into anatomical loci, each with distinct functional contributions to sensation, perception, and behavior (*Santiago Ram´on y Cajal, 1899*; *Brodmann, 1909*; *Penfield and Jasper, 1951*). In other words, information carried by neighboring neurons is likely to be similar in content, be it visual or interoceptive. However, much of our understanding of the neural code is derived from experimental designs that manipulate stimuli or measure behavior. This approach leaves innumerable other forms of biologically relevant features as potentially latent variables, such as reliable identifying information about the neuron itself. A complete understanding of these latent variables is essential for a complete understanding of a neuron's role in the brain.

The null hypothesis is that the impact of anatomy on a neuron's activity is either nonexistent or unremarkable. This is supported by three observations at the level of models. First, from a computational perspective, neurons' outputs primarily reflect their inputs along with noise (*Softky and Koch, 1993*; *Shadlen and Newsome, 1994*; *London et al., 2010*). Second, artificial recurrent neural networks, which also perform input integration, can emulate patterns of neural circuit activity and functions central to biology, including motor control (*Pandarinath et al., 2018*) and visual processing (*Sawant et al., 2022*). Yet, such networks do not require the formalized structure that is a hallmark of brain organization. Finally, in deep neural networks, anatomy is only relevant insofar as progressively deeper layers are functionally distinct. Taken together, complex information processing is achievable with neither a strict concept of anatomy nor a computational encoding of anatomy.

However, there are three principal reasons to justify asking if neurons reliably embed their anatomical location in their spiking. First, recent work suggests that, in addition to stimulus information, neurons transmit their genetic identity (*Schneider et al., 2023*). Second, in contrast to artificial systems, there are conceptual reasons why a brain might benefit from a reliable neuron-level code for anatomy. For example, anatomical information is presumably crucial during neurodevelopment (*Patel and Poo, 1982*). Likewise, such information could be valuable when parsing inputs during multimodal integration (*Gütig and Sompolinsky, 2006*). Finally, within a species, brain anatomy is highly stereotyped. Thus, it stands to reason that, should a neuron's anatomy be embedded in its spike train, this embedding might generalize across individuals.

Historically, the classification of brain regions has utilized anatomical landmarks, functional outputs, and, more recently, genetic markers, creating a comprehensive map of neural diversity. There is evidence that neural activity may vary by region, although such observations are restricted to population-level statistics (*Siegle et al., 2021*). Specifically, patterns of neural activity evolve gradually across the anatomical landscape, influenced by gradients of synaptic weight (*Elston, 2007*; *Chaudhuri et al., 2015*), the autocorrelation timescale (*Murray et al., 2014*), and connectivity patterns (*Maunsell and van Essen, 1983*; *Gămănuţ et al., 2018*; *Harris et al., 2019*). While indicative of the brain's complex architecture, these observations comprise subtle, statistical differences between populations rather than stark, unique signatures that could be used to classify an individual neuron with any confidence. Thus, it is unclear whether different computational rules have evolved in functionally distinct circuits. Alternatively, subtle statistical changes could reflect methodological limitations to capturing the full spectrum of activity patterns that define distinct brain regions. Should robust, circuit-specific rules exist, their description would enable a more precise understanding of how regional variations contribute to the broader neural code and sharpen our understanding of the brain's computational organization.

Ultimately, the question can be distilled as, what is the minimum spatial scale at which neural activity patterns are reliably structured by the region of origin? Such patterns, should they exist, are most meaningful when identified at the level of single neurons. However, given the variance observed in any number of neuronal features—from tuning properties, to genetic cell type, to connectivity — it is unlikely that anatomy reliably determines neural activity at an obvious level, if at all. To address this, we employed a supervised machine learning approach to analyze publicly available datasets of high-density, multi-region, single-unit recordings in awake and behaving mice. To specifically evaluate whether anatomical information is embedded in the neuronal code, we examined the timing of spikes generated by well-isolated single units. We examined the possibility of anatomically defined computational rules at four levels: (1) large-scale brain regions (hippocampus, midbrain, thalamus, and

visual isocortex), (2) hippocampal structures (CA1, CA3, dentate gyrus, prosubiculum, and subiculum), (3) thalamic structures (ethmoid nucleus, dorsal lateral geniculate, lateral posterior nucleus, ventral medial geniculate, posterior complex, suprageniculate nucleus, ventral posteromedial nucleus), and (4) visual cortical structures (primary, anterolateral, anteromedial, lateral, posteromedial, rostrolateral). We find that traditional measures of neuronal activity (e.g. firing rate) alone are unable to reliably distinguish anatomical location. In contrast, using a multi-layer perceptron (MLP), we reveal that information about brain regions and structure is recoverable from more complete representations of single unit spiking (e.g. interspike interval distribution). Further, we demonstrate that learning the computational anatomical rules in one animal is sufficient to decode another, both within and across distinct research groups. These observations suggest a conserved code for anatomical origin in the output of individual neurons, and that, in modern datasets that span multiple regions, electrode localization can be assisted by spike timing.

## Results

### Dataset and inclusion criteria

To evaluate whether individual neurons embed reliable information about their structural localization in their spike trains, we required consistently reproduced recordings of large numbers of single units distributed throughout numerous cortical and subcortical structures. Further, we reasoned that candidate recordings should involve only awake and behaving animals. Finally, while highly restricted and repetitive stimuli are frequently leveraged to average out noise (*Millman et al., 2020*), this approach increases the likelihood of overfitting in our case. Thus, we only considered datasets that contained diverse stimuli as well as spontaneous activity. Two open datasets from the Allen Institute meet these criteria, specifically (1) Brain Observatory and (2) Functional Connectivity (*Siegle et al., 2021*). These datasets comprise tens of thousands of neurons recorded with high-density silicon probes (Neuropixels) in a total of N=58 mice (BO N=32, FC N=26). We used the selections of drifting gratings and the naturalistic movie found in the Brain Observatory dataset, which were distinct from those in the Functional Connectivity dataset. Additionally, we used recordings of spontaneous activity during blank screen presentations from both the Brain Observatory and Functional Connectivity datasets. Raw data were spike sorted at the Allen Institute. Because information is primarily encoded by the firing rate or the timing of spiking and not waveforms (etc)., our studies involve only the timestamps of individual spikes from well-isolated units (*Figure 1A*). These units were filtered according to objective quality metrics such as ISI violations, presence ratio, and amplitude cutoff (see Methods). At the largest spatial division, neurons were assigned to four distinct brain regions: hippocampus, midbrain, thalamus, and visual cortices. Within regions, neurons were assigned to fine-grained structures, for example, CA1 hippocampus, lateral posterior thalamus, and anteromedial visual cortex (*Figure 1B*). Note that midbrain neurons were not further classified by structure due to the relatively low number of neurons recorded there (to be considered at the structure level, we required a minimum of n=150 neurons). We tested the possibility of a computational embedding of anatomical location at two levels of generalizability. In the transductive approach, all neurons from all animals were merged before splitting into a training set and a testing set. This arrangement preserves the capacity of a model to learn some within-animal features. In contrast, for the inductive approach, model training is performed on all neurons from a set of animals, and model testing is performed on neurons from entirely withheld animals. In this case, any successful learning must, by definition, be generalizable across animals (*Figure 1A*).

### Exploratory trends in spiking by region and structure across the population

Many studies have highlighted statistical differences in the activity of populations of neurons as a function of brain region and structure (*Murray et al., 2014*; *Shinomoto et al., 2009*; *Mochizuki et al., 2016*; *Wang, 2022*). At face value, this raises the possibility that anatomy may have a powerful role in shaping spiking patterns. However, the existence of population-level differences does not necessarily imply that the anatomical location of an individual neuron can be reliably inferred from its spike train alone; especially in large datasets, even subtle regional differences in spiking may be significant. Put simply, statistically separable populations can exhibit extensive overlap (*Hastie et al., 2009*). Prior

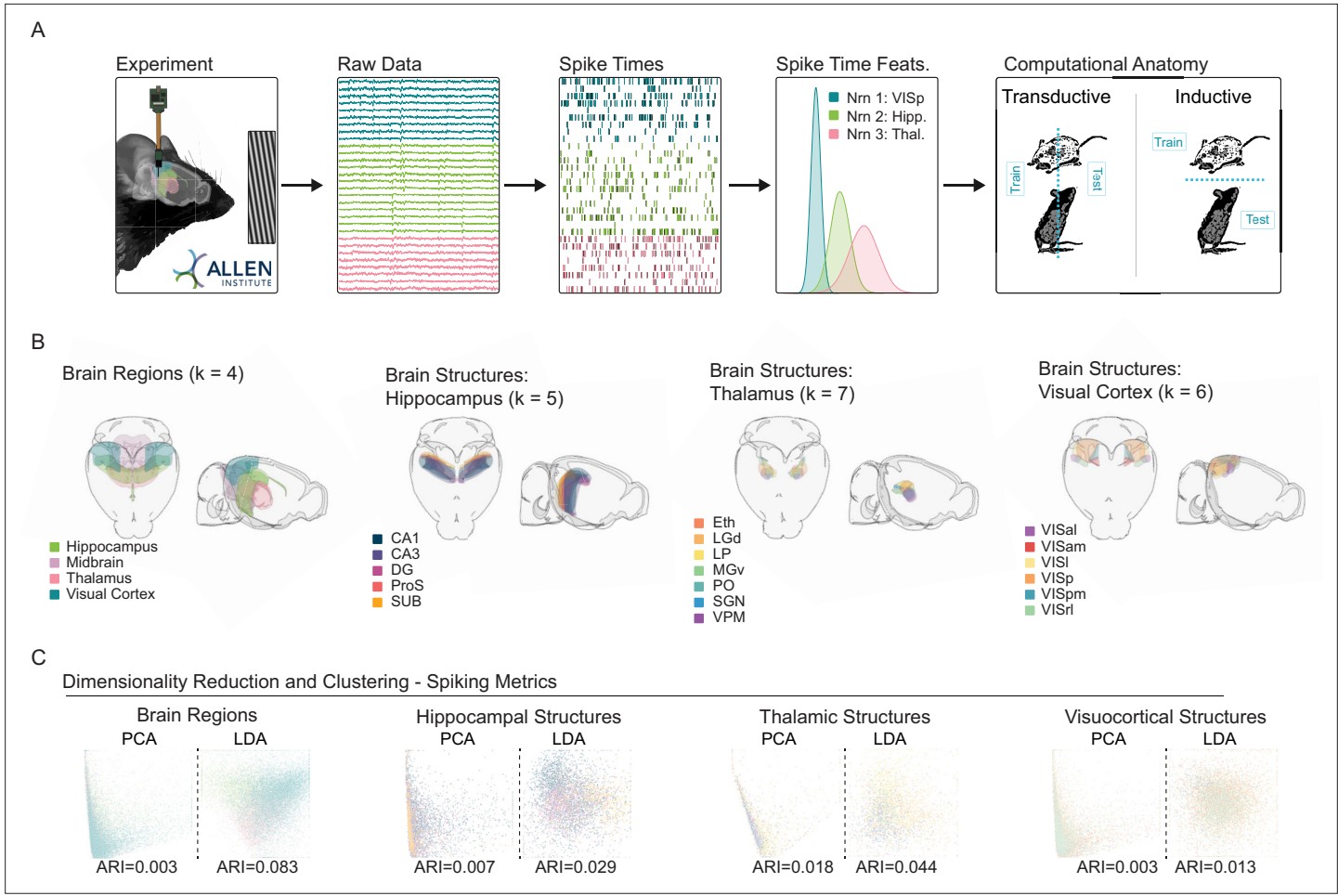

**Figure 1.** Dimensionality reduction suggests a limited relationship between neuroanatomy and spike timing. (**A**) Experimental pipeline. Left to right: Recording, raw data, extracted spike times, spiking time features (e.g., rates, CV), and model training protocols. The Allen Institute Visual Coding dataset comprises high-density silicon extracellular recordings that span multiple brain regions and structures. During recording, mice were headfixed and presented with multiple visual stimuli, including drifting gratings. For supervised experiments, classifiers were either transductive—all neurons from all animals were mixed and divided into train and test sets, or inductive—train and test sets were divided at the level of the animal. (**B**) Brain regions and structures included in our analyses. (Left to right) Brain Regions: Hippocampus, Midbrain, Thalamus and Visual Cortex. Hippocampal Structures: CA1, CA3, Dentate Gyrus (DG), Prosubiculum (ProS), Subiculum (SUB). Thalamic Structures: Ethmoid Nucleus (Eth), Dorsal Lateral Geniculate (LGd), Lateral Posterior Nucleus (LP), Ventral Medial Geniculate (MGv), Posterior Complex (PO), Suprageniculate Nucleus (SGN), Ventral Posteromedial Nucleus (VPM). Visuocortical Structures: Anterolateral (VISal), Anteromedial (VISam), Lateral (VISl), Primary (VISp), Posteromedial (VISpm), Rostrolateral (VISrl). (**C**) PCA/LDA plot of units recorded in each set of regions/structures (values capped to 1st and 99th percentiles for visualization- see Exploratory Visualization section of Methods). For each unit, 14 spiking metrics (see methods) describe the spike train, which is then placed in a 2D scatterplot. Color scheme follows 1B. Adjusted Rand Index (ARI) of clustering with respect to ground truth labels of regions and structures is printed for each plot.

The online version of this article includes the following figure supplement(s) for figure 1:

**Figure supplement 1.** Exploratory visualization of spiking activity.

to considering patterns embedded in individual spike trains, we asked whether and to what extent anatomical location explains the diversity of spiking patterns when considered across the entire population of included neurons (n=18,691 neurons from N=32 mice). Specifically, we applied exploratory visualizations to observe the dominant trends in data with respect to both overall variance (unsupervised) and separability (supervised).

We examined three distinct representations of spiking activity: (1) a previously described collection of 14 well-established spiking metrics (e.g. firing rate, coefficient of variation, etc.; *Schneider et al., 2023*), (2) the distribution of interspike intervals (ISIs; 0–3 s, 100 uniform bins), and (3) the trial averaged PSTHs for each drifting grating condition (the collection of gratings included 8 orientations and 5 frequencies, PSTH was calculated in 100 msec bins across a 3 s trial). The Principal Components

Analysis (PCA) and Linear Discriminant Analysis (LDA) scatter plots suggest that, depending on the combination of analytical method and included features, there are often subtle influences of brain region and structure in these dimensionally reduced spaces (*Figure 1C*, *Figure 1—figure supplement 1*). However, across all 24 combinations of anatomical tasks, analytical method, and examined features, there was not a single example of a prominent clustering organization aligned with brain structures or brain regions, as indicated by the adjusted rand index. These data may suggest that, examined as large populations, various features of neuronal spike timing may differ between regions, but that such differences are neither distinct nor robust. However, in all cases, greater alignment of labels and clusters was observed in supervised LDA, suggesting the potential of supervised analyses to extract targeted insights about structure and anatomy (*Figure 1—figure supplement 1*).

## Separability of brain structures using established spiking metrics

Conservatively, it is possible that our unsupervised analysis (*Figure 1*) reflects a true limit of the separability of structures based on spiking activity. However, by definition, unsupervised approaches reflect the most prominent trends (e.g. the largest sources of variance, in the case of PCA) in neuronal activity, which may or may not be related to anatomical location. As a result, it is possible that an analysis whose effectiveness is defined by the separability of region and structure may reveal more reliable anatomical rules. The more apparent separability in our LDA suggests the potential of such supervised analyses. To quantify this, we asked whether, given the spiking activity of a single unit, its region and/or structure could be successfully predicted by a machine-learning model.

Broadly, in our initial supervised experiments, we pooled units from all animals. Pooled neurons were divided into non-overlapping sets for training (60%), validation (20%), and testing (20%). Crucially, equal proportions of each animal's neurons were maintained in each set. This architecture allowed us to seek patterns across all animals that generalize to unseen neurons from the same set of animals (*Figure 1A* Computational Anatomy). Here, borrowed from machine learning, we use the term *transductive* to describe this approach which can transfer patterns learned from neighboring neurons to unseen ones (*Pan and Yang, 2010*; *Vapnik, 1998*).

Model performance is quantified with balanced accuracy, which is the average prediction accuracy for each class (i.e. region or structure) (*Brodersen et al., 2010*). Error is ± SEM, and chance is 1/number of classes. To clearly indicate how models make errors between classes, we use confusion matrices. Confusion matrices are square heat maps where rows represent true class labels and columns represent predicted class labels. Diagonal entries indicate the proportion of units correctly predicted for their region or structure, while off-diagonal entries reflect instances of the model's 'confusion'. An effective model will exhibit high balanced accuracy and a strong diagonal structure in the corresponding confusion matrix.

We first passed standard measures of neuronal activity to a simple model. Specifically, we trained logistic regressions to predict a unit's region/structure from 14 statistical measures that comprised three categories: (1) standard statistics, such as mean or maximum firing rate, (2) measures of local variance, such as CV2, a temporally constrained alternative to the coefficient of variation (CV; *Holt et al., 1996*), and (3) spectral power, such as the delta band (0.1–4 Hz). Models were trained and tested in each of four prediction tasks: (1) brain regions, (2) hippocampal structures, (3) thalamic structures, and (4) visuocortical structures (*Figure 2A–D*). In each of these tasks, we evaluated the effectiveness of the 14 features individually as well as in combination to predict the anatomical localization of individual neurons (*Figure 2*).

In all four tasks, the majority of individual spiking metrics supported weak but above-chance classification (*Figure 2A–D*). In contrast to brain regions and hippocampal structures, the ability of the 14 metrics to reliably indicate different visuocortical structures was notably low. Ordered by their classification performance (*Figure 2A–D*, x-axes), the arrangement of spiking metrics was highly task dependent. We examined the correlation of metric ordering — the order of classification effectiveness for the spiking metrics, from greatest to least — between the region task and the three structure tasks. We found a range of correlation from 0.03 to 0.73 (Spearman Rank Correlations- regions/VC: $p = 0.0336$, regions/TH: $p = 0.0814$, regions/HC: $p = 0.7253$), suggesting that the characteristics of the spike train that best discriminate regions are in some cases distinct from the characteristics used to discriminate structure. In combination, the 14 metrics substantially increased the overall balanced accuracy for each of the four tasks, although the visuocortical task remained challenging

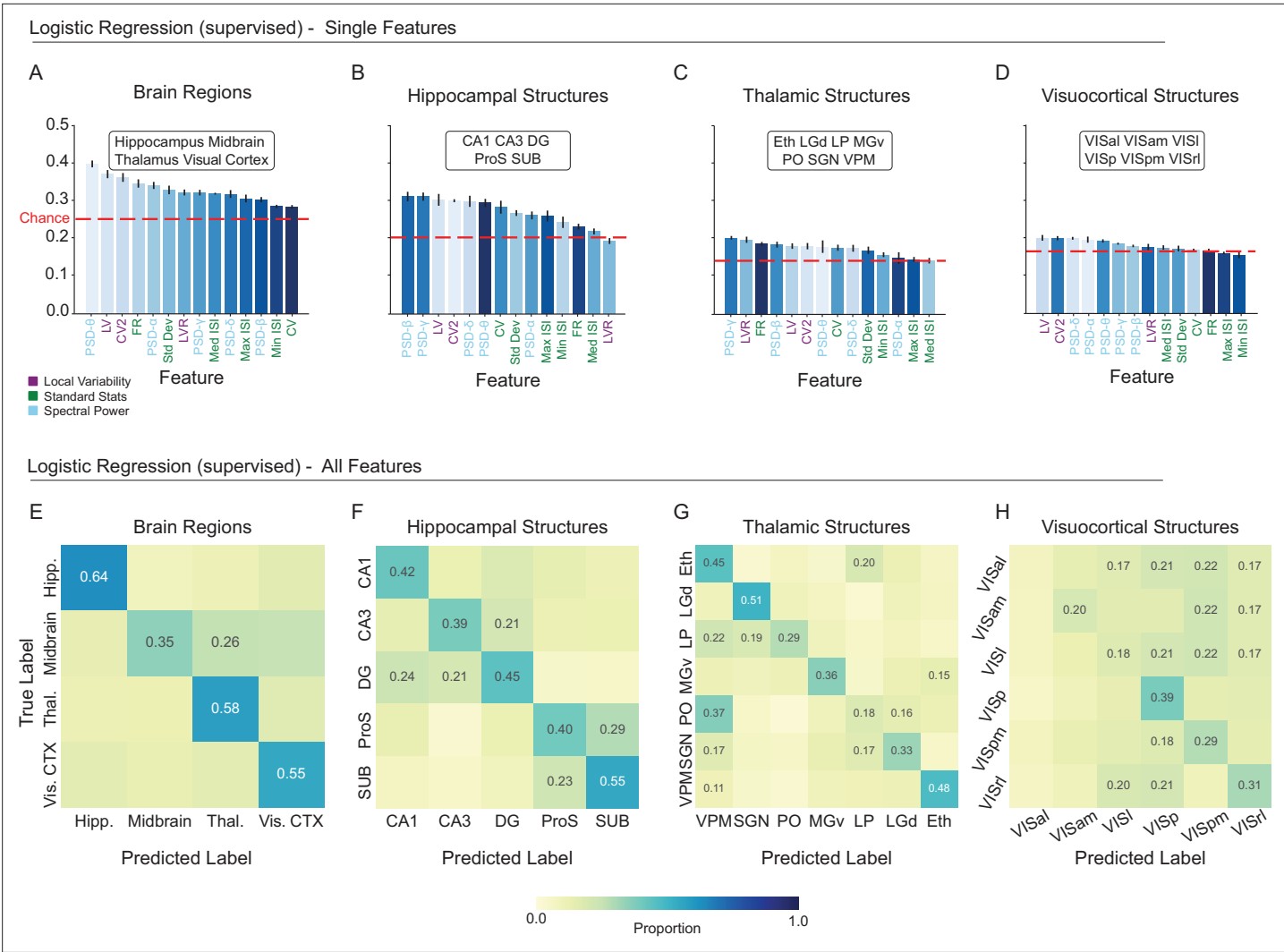

**Figure 2.** A linear classifier can learn to predict single neuron location based on standard spiking metrics. (**A-D**) Balanced accuracy of logistic regression models trained to predict the anatomical location of single units based on each of 14 individual spiking metrics: Coefficient of Variation 2 (CV2), Local Variation (LV), Revised Local Variation (LVR), Mean Firing Rate (FR), Standard Deviation (Std Dev) of interspike intervals (ISIs), Coefficient of Variation (CV), Minimum ISI (Min ISI), Median ISI (Med ISI), Maximum ISI (Max ISI), power spectral density (PSD)-δ (Delta Band 0.1–4 Hz), PSD- $\theta$ (Theta Band 4–8 Hz), PSD-α (Alpha Band 8–12 Hz), PSD-β (Beta Band 12–40 Hz), PSD-γ (Gamma Band 40–100 Hz). Balanced accuracy expected by chance varies by task and is indicated by the dashed red line. Features on the x-axis are ordered by performance. Feature (bar) colors are assigned by the ordering in A (brain region task) and maintained for the structure tasks. Error bars represent the standard error of the mean (SEM) across five splits. (**B–D**). This shows the extent to which individual features maintain their relative importance across tasks. (**E-H**) Confusion matrices from logistic regression models trained to predict unit location from the combination of all 14 spiking metrics. Each confusion matrix shows the average of 5 train/test splits of the data. The proportion is printed only in cells where the proportion was greater than chance level (1/number of classes). Balanced accuracy for each task: Brain Regions = 52.91 ± 1.24; Hippocampal Structures = 44.10 ± 1.99; Thalamic Structures = 37.14 ± 2.57; Visuocortical Structures = 24.09 ± 1.46 (error is the SEM across 5 splits).

(*Figure 2E–H*; Brain Regions Balanced Accuracy = 52.91 ± 1.24; Hippocampal Structures = 44.10 ± 1.99; Thalamic Structures = 37.14 ± 2.57; Visuocortical Structures = 24.09 ± 1.46). Perhaps unsurprisingly, the classification of regions was more effective than the structures within them. Taken together, these results demonstrate an intermediate divisibility of regions and structures by logistic regression based on spiking metrics. Given that spiking metrics were predetermined and that logistic regression can only learn linearly discriminable features, it is possible that more robust anatomical information could emerge from the spike train with more complex learned representations.

# Flexible, non-linear embedding of anatomical information in neuronal spiking

The 14 spiking metrics were selected based on prior literature (*Schneider et al., 2023*). However, spike timing can vary along an enormous number of axes, many of which are not indicated by preselected features. To test the possibility that non-parametric, data-driven representations of a neuron's spiking might contain additional, valuable information regarding anatomy, we considered the full collection of each neuron's interspike intervals (ISIs), that is the time between two consecutive spikes. Prior work suggests the increased complexity of the ISI distribution is best captured by a nonlinear classification model (*Schneider et al., 2023*). Thus, we employed a multilayer perceptron (MLP). MLPs are a relatively simple class of feedforward artificial neural networks. Briefly, an MLP consists of an input layer, one or more hidden layers, and an output layer. Each layer is composed of interconnected nodes (artificial neurons) that process and transmit information via weighted connections and nonlinear activation functions. This enables MLPs to learn and model complex relationships between input features and output targets.

Broadly, MLPs were more successful in extracting anatomical information from spike trains than logistic regressions trained on spiking metrics (*Figure 3A*). Specifically, MLPs were trained transductively (which enables transfer of patterns learned from neurons to unseen neighbors) on each of three arrangements of spike times: (1) the entire distribution of ISIs generated during the presentation of drifting gratings (0–3 s in 100 uniform bins), (2) each neuron's average PSTH across all stimuli—essentially describing a cell's mean response to stimuli, and (3) the concatenated mean PSTHs for each of the 40 combinations of drifting grating orientation and frequency. Across all four classification tasks (brain regions and three sets of structures), MLPs trained on the ISI distribution and concatenated PSTHs outperformed logistic regression models. Conversely, MLPs trained on the average PSTH (averaged across all drifting gratings without respect to orientation or frequency) exhibited reduced balanced accuracy (*Figure 3*). Together, these data suggest that the embedding of anatomical information in a neuron's activity is recoverable when some form of precise information about spike time is available, and that anatomical information is degraded by averaging.

Due to their reliance on hidden layers, MLPs are prone to the black-box effect; deep models often offer limited insight into the underlying mechanisms of their performance. To elucidate the features of ISI distributions and PSTHs that MLPs use to infer a neuron's anatomical location, we strategically manipulated three features of our data: (1) we varied input duration, (2) we permuted specific bands of the ISI distribution, and (3) we permuted neuronal responses to specific drifting grating orientation/frequency combinations.

We passed progressively reduced lengths of input data, ranging from 1800s down to 0.1 s, into the fully trained model and evaluated its sensitivity (true positive rate). This approach allowed us to assess the timescale of the anatomically informative patterns learned by the MLP. If short intervals of data reliably indicated anatomy, it would suggest that models learn to detect acute and reliable signatures. Conversely, if model sensitivity increased monotonically with time, it would indicate that anatomical information is temporally distributed and cumulative. Our manipulation of the test set duration supported the latter hypothesis; sensitivity for all four brain regions increased as input data duration increased. In all conditions, the slope of the input duration versus sensitivity line was still positive at 1800s (*Figure 3B*). Because the training data were of similar duration, this could be explained by either of two possibilities. First, the signal is relatively short, but noisy—in this case, extended sampling will increase reliability. Second, the anatomical signal is, itself, distributed over time scales of tens to hundreds of seconds.

ISIs vary extensively, from the biophysical minimum of the absolute refractory period to many seconds of silence. To test whether anatomical information might be enriched in specific temporal ISI windows, we assembled each neuron's entire ISI distribution (0–3 s in 10ms bins) and selectively permuted subsets of intervals before passing the distribution to the trained MLP for classification. We shuffled ISI counts across neurons within specific temporal windows (e.g. ISIs between 50 and 60ms). This approach identified regions of the ISI distribution informative for classification. Consistent with temporal tuning, certain subsets of ISIs were more important than others for MLP performance, with the critical time windows varying by brain region (*Figure 3C*). For instance, very fast ISIs (<~150ms) were particularly valuable for identifying midbrain neurons, while ISIs of 200–600ms made an outsized contribution to visual cortical classification.

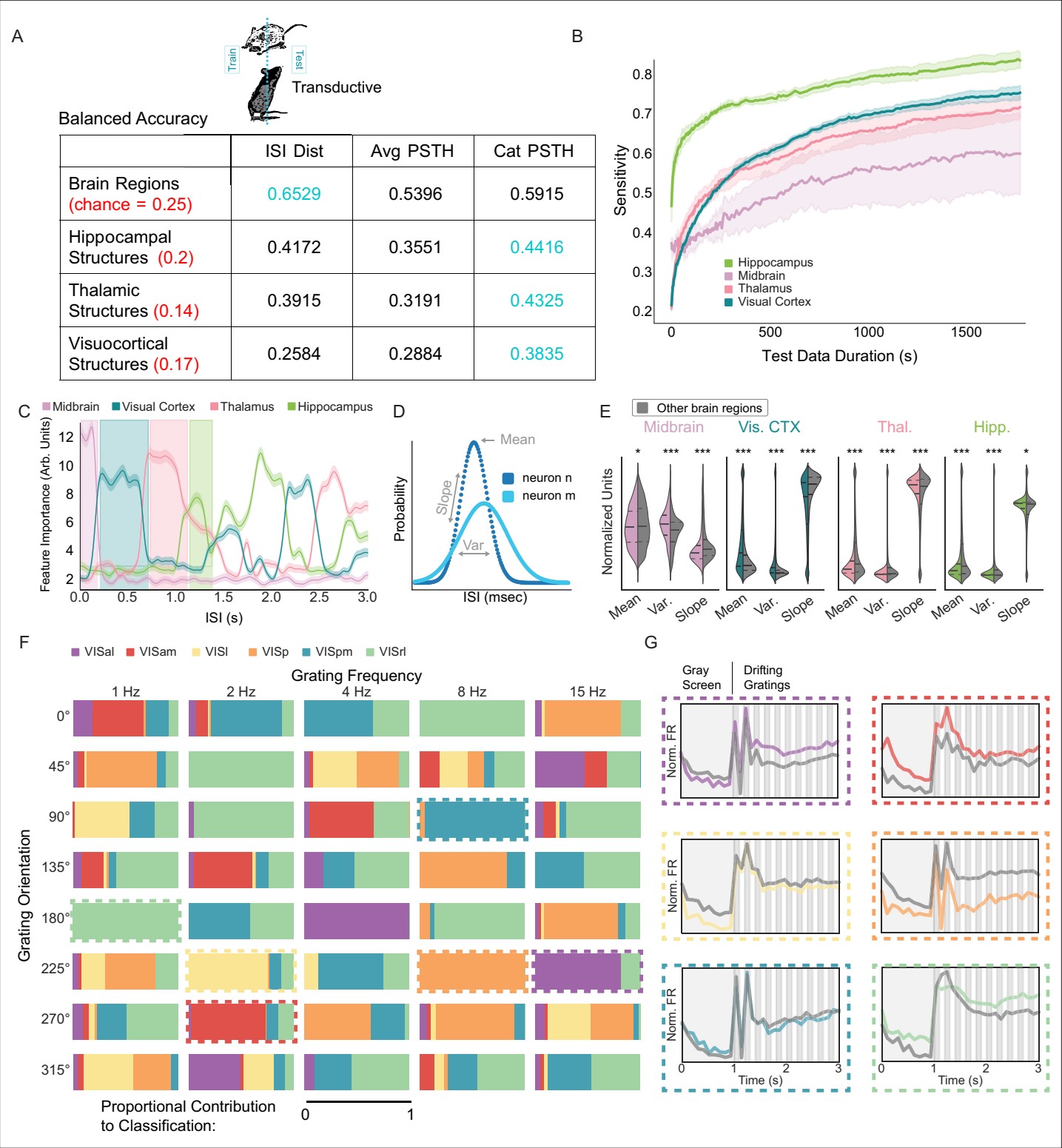

**Figure 3.** Anatomical information in spike trains is captured by nonlinear models that learn patterns in ISIs and stimulus-specific responses. (**A**) Average balanced accuracy of transductive multi-layer perceptrons (MLPs) in classifying unit location in each task (rows) based on three representations of the spike train (columns). Chance is red, and peak balanced accuracy is blue. ISI dist—full distribution of ISIs; Avg PSTH—mean peristimulus time histogram across all trials; Cat PSTH—concatenation of PSTHs from all 40 stimuli (see Methods). (**B**) MLP sensitivity as a function of test data duration. Model was trained normally and tested on varying amounts of data from each of the four brain regions. (**C**) Feature importance from models that classified anatomy based on ISI dist. Features are ISI ranges between 0 and 3 s in 10 ms bins. 5 splits and 100 iterations. Error is ± 1 SEM across 100 shuffles. High value

*Figure 3 continued on next page*

*Figure 3 continued*

ranges for each region are highlighted. (**D**) Illustration of ISI mean, slope, and variance. (**E**) Regional distributions of mean, slope, and variance within the highlighted range (in **C**) compared to averages from all other regions within that range (gray). Multiple comparisons corrected t-tests: $p >= 0.05$, *: $p < 0.05$, **: $p < 0.01$, ***: $p < 0.001$. Corresponding effect sizes as absolute value of Cohen's d (left-to-right): 0.0697, 0.4391, 0.2080, 0.4029, 0.3417, 0.5682, 0.0958, 0.0621, 0.0697, 0.2521, 0.1000, 0.0382. Visuocortical structure information is enriched in subsets of stimuli. Each rectangle represents one of the 40 stimulus parameter combinations. Colors correspond to visuocortical structures, and the area of color shows the relative importance of that stimulus to model classification of the given structure. Cat PSTH model. Dashed boxes—exemplar stimulus conditions. (**G**) PSTHs corresponding to structure/stimulus pair indicated in (**F**). Specific structure PSTH shown in color. Average PSTH of all other structures shown in gray.

The online version of this article includes the following figure supplement(s) for figure 3:

**Figure supplement 1.** Unsmoothed transductive classification: confusion matrices.

**Figure supplement 2.** MLP-based model sensitivity as a function of input duration.

**Figure supplement 3.** Interpretation of hippocampal and thalamic MLPs.

To gain insight into how regional information could be enriched within these ISI ranges, we examined ISI distribution properties within each identified range, contrasting the region preferentially embedded in that range with all others. Across every region and range, there were subtle but statistically significant differences in the mean, variance, and slope of distributions (*Figure 3D and E*). This suggests the presence of coarse features in regionally relevant spiking patterns but does not rule out the contribution of a combination of multiple ISI ranges for reliable embedding.

Brain region, the coarsest anatomical classification task, was most effectively extracted from broad ISI distributions (65.29 ± 0.97% Balanced Accuracy vs. 59.15 ± 0.83% concatenated PSTH vs. 52.91 ± 1.23% in spike metrics-based logistic regression, chance = 25%; *Figure 3A*). In contrast, the discriminability of smaller and more numerous structures was greatest when MLPs were trained on detailed spike timing information (i.e. concatenated PSTHs; *Figure 3A*, *Figure 3—figure supplement 1*). Somewhat intuitively, the most robust embedding of visual cortical structures was obtained in conjunction with visual stimulus information (38.35 ± 3.19% Balanced Accuracy vs. 25.84 ± 1.35% ISI distribution vs. 24.09 ± 1.46% in spike metrics-based logistic regression, chance = 17%; *Figure 3A*, *Figure 3—figure supplement 1*).

We next asked whether structural information might be conveyed by differing responses to specific stimuli. Analogous to our methods for ISIs, we shuffled the values of particular PSTHs (with specified orientation and frequency) across neurons prior to passing PSTHs to the trained MLP for structure classification. We found that information about each of the visuocortical structures was embedded in the response to multiple stimulus variants. In other words, the MLP learning of visuocortical structure was not characterized by responses to a single set of stimulus parameters (*Figure 3F*). However, despite the distributed nature of structured information, some stimuli carried more predictive value than others (*Figure 3G*). Interestingly, despite PSTHs being defined by the presentation of visual stimuli, similar improvements were observed in hippocampal and thalamic structure tasks (Hippocampal structures: 44.16 ± 2.48% Balanced Accuracy vs. 41.72 ± 2.36% ISI distribution vs. 44.10 ± 1.99% in spike metrics-based logistic regression, chance = 20%; Thalamic structures: 43.25 ± 3.68% Balanced Accuracy vs. 39.15 ± 2.01% ISI distribution vs. 37.14 ± 2.57% in spike metrics-based logistic regression, chance = 14%; *Figure 3A*, *Figure 3—figure supplement 1*). This finding is noteworthy because, with the exception of the LGN, none of the constituent structures are primarily associated with visual processing.

## Anatomical codes generalize across animals and conditions

Our results demonstrate that anatomical information is embedded in the spike trains of single neurons. However, thus far, our results employ transductive models (which can learn patterns for unseen neurons from their immediate neighbors in the same animal). This raises the possibility that MLPs learned to identify animal-specific spiking patterns that do not generalize across individuals. As an example, if hippocampal neurons from animal *n* happen to be characterized by a specific rhythm, it is reasonable to assume that their neighbors, who happen to be withheld for the test set, exhibit similar activity. Transductive models are agnostic to whether such a pattern is anatomically meaningful and simply a one-off correlation that happens to be localized. To disambiguate these possibilities, we adopted an inductive approach in which the train and test sets are divided at the level of the entire

animal. In other words, patterns are learned in one set of animals and tested on another group of animals that is entirely new to the model (*Figure 1A*).

We trained and tested inductive models on the four classification tasks — brain regions, hippocampal structures, thalamic structures, and visuocortical structures – across two input conditions, general ISI distributions and concatenated PSTHs. The null hypothesis is that anatomical information generalizes within the animal but not to new animals. In support of the alternate—that anatomical embedding is a universal feature in the spike train–inductive models performed significantly above chance in seven out of eight conditions and were statistically indistinguishable from the performance of transductive models for all four ISI-based tasks. In the four concatenated PSTH tasks, inductive models exhibited significantly lower balanced accuracy than transductive models, although they were still above chance in all but the hippocampal structures task (*Figure 4A*). These results suggest that stimulus-dependent representations of anatomical location are predominately specific to the animal, while ISI distribution-based anatomical information is general across animals.

## Single neuron classification errors can be corrected by population context

While the ISI-based inductive models demonstrate that generalizable anatomical information is carried in single unit spike trains, classification is imperfect. Note that the model classifies individual neurons via a winner-take-all approach. Even for correctly labeled neurons, the probability for the chosen class only needs to narrowly exceed the probability of other options. We next asked to what extent such uncertainty could be mitigated if the consensus was taken across a group of neurons, analogous to how the brain processes noisy information (*Riehle et al., 1997*; *Maass, 2000*; *Paz et al., 2016*). One possibility is that uncertainty is driven by shared noise amongst neighboring cells. Errors of this form could be amplified by a consensus vote. Alternatively, if incorrect classifications are stochastically distributed across a recording array, errors should be trivially correctable by considering the surrounding ensemble—a stray CA1 neuron should not be detected in the middle of CA3, for example. To test this, we added a second step following single unit classification. This step comprised the use of a Gaussian kernel to smooth anatomical probabilities across neurons recorded on neighboring electrodes.

In support of the stochastic error hypothesis, smoothing dramatically increased the balanced accuracy of MLPs in all four tasks. The brain region identification task improved from 65.10 ± 1.77%–89.47 ± 2.98% compared to non-smoothed inductive models. The smoothed regional prediction probabilities across an individual probe from an example animal are shown in *Figure 4B*. Similarly, the hippocampal task improved from 35.89 ± 1.42%–51.01 ± 4.50%, the thalamic task improved from 32.79 ± 2.37%–53.21 ± 7.59%, and the visuocortical task improved from 25.52 ± 0.73%–38.48 ± 3.31% (*Figure 4—figure supplement 1*, *Figure 4—figure supplement 2*). This suggests that erroneously labeled neurons do not share anatomically relevant spike patterns with nearby cells, and thus are amenable to consensus-based correction.

To this point, MLPs were trained and tested only on neurons related to an individual task. For example, MLPs trained on visuocortical structures were never exposed to thalamic neurons. To evaluate inductive models across the full set of regional and structural comparisons while capitalizing on smoothing of errors, we trained a hierarchical model that combined smoothed brain region classification with subsequent smoothed structure identification. Across all 19 labels (18 structures and midbrain), the hierarchical inductive model achieved a balanced accuracy of 46.91 ±1.90% (*Figure 4C*). An observable effect of a hierarchical approach is that, because smoothed brain region classification is so effective, errors generally occur only between similar structures (VISl vs. VISal) rather than highly divergent ones (VISl vs. CA1). Here, only 5 out of the total 250 possible cross-regional errors occurred above chance (*Figure 4C*). These data demonstrate that, with smoothing, a hierarchical model can extract surprisingly effective anatomical structure information from single neuron spike timing that generalizes to unseen animals. This further suggests that there is a latent neural code for anatomical location embedded within the spike train, a feature that could be practically applied to determining the brain region of a recording electrode without the need for post-hoc histology. While significantly above chance, the structure-level model still lacks the accuracy for immediate practical application. However, it is highly likely that the incorporation of datasets with diverse multi-modal features and alternative regions from other research groups will increase the accuracy of such a model. In addition, a computational approach can be combined with other methods of anatomical reconstruction.

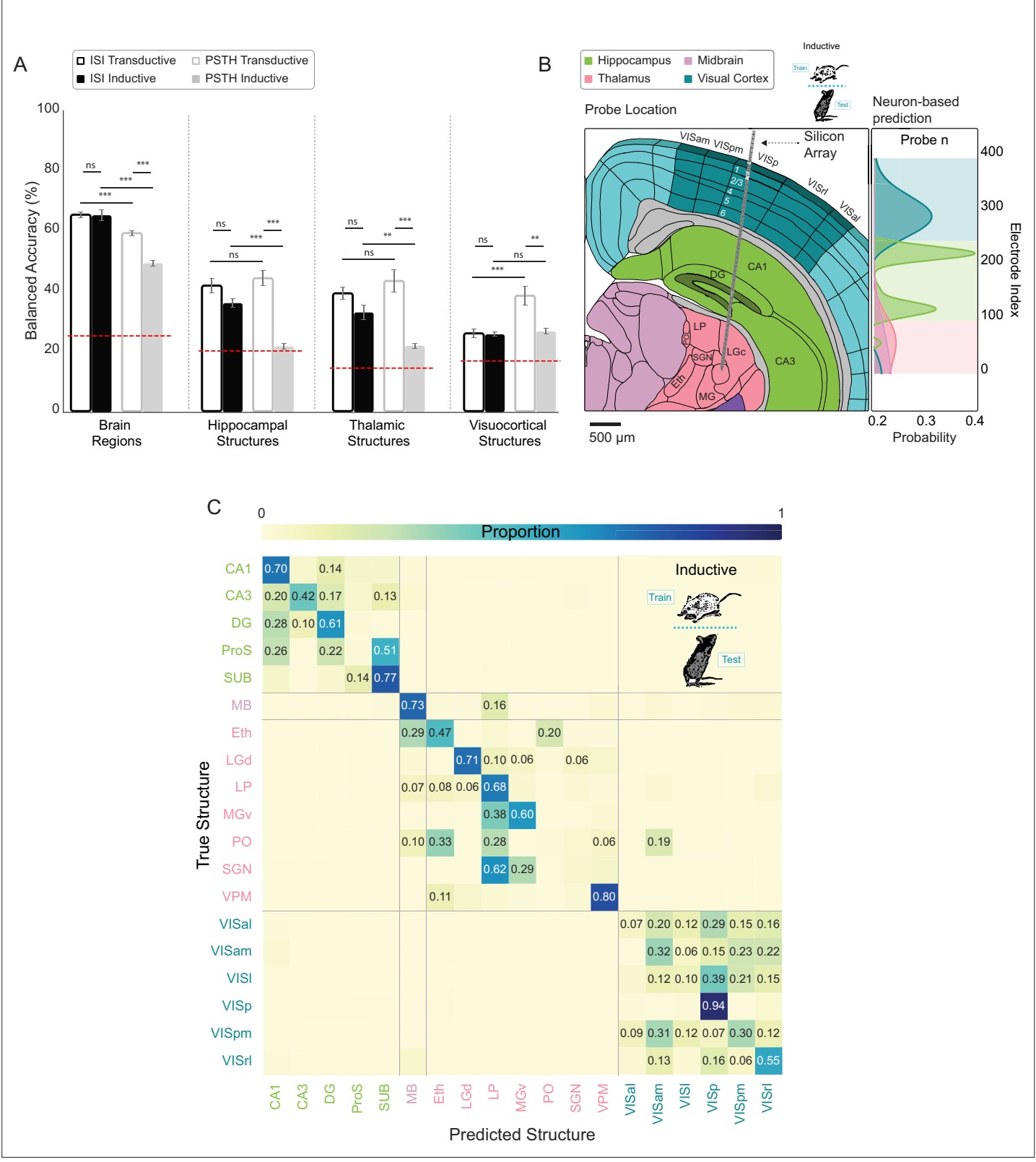

**Figure 4.** Spike time features that predict anatomy generalize across animals. (**A**) Mean balanced accuracy (± SEM across 5 splits) of multi-layer perceptrons (MLPs) trained on ISI distributions and concatenated PSTHs in both inductive (withhold entire animals for testing) and transductive splits. Chance (red dashed line) varies by task. Linear mixed effects regression (ns/not significant: $p >= 0.05$, *: $p < 0.05$, **: $p < 0.01$, ***: $p < 0.001$). (**B**) Left: Illustration of an example implanted silicon array spanning isocortex, hippocampus, and thalamus. Right: Brain region probability for the example

*Figure 4 continued on next page*

*Figure 4 continued*

implant shown on the left calculated by smoothing across neuron-based classifications. Colored background shows the consensus prediction as a function of neuron location (electrode number). (**C**) Confusion matrix resulting from hierarchical (region then structure) inductive classification with smoothing. Matrix cells with proportion less than chance (1/number of classes) contain no text. Average balanced accuracy after smoothing (by task): Brain Regions = 89.47 ± 2.98%; Hippocampal Structures = 51.01 ± 4.50%; Thalamic Structures = 53.21 ± 7.59%; Visuocortical Structures = 38.48 ± 3.31% (error is the SEM across 5 splits). Overall balanced accuracy: 46.91 ±1.90%.

The online version of this article includes the following figure supplement(s) for figure 4:

**Figure supplement 1.** Unsmoothed inductive classification: confusion matrices.

**Figure supplement 2.** Smoothed inductive classification: confusion matrices.

## Spike train embedding of visual superstructure and cortical layer

Even with smoothing, visuocortical structures remained the most challenging to classify. There are two possible explanations for this. First, it may be that there are truly no consistent differences in the organization of single unit spiking across visuocortical structures. Alternatively, there are true differences but our MLP-based approach fails to learn them. Although there are clear cytoarchitectural boundaries between primary visual cortex (VISp) and secondary visual areas, the differentiation of secondary visual areas is functionally determined (*Wang and Burkhalter, 2007*; *Glickfeld and Olsen, 2017*). We asked whether broadly defined superstructures, VISp and VISs—a combination of all secondary areas (VISal, VISam, VISl, VISpm, and VISrl)–increases the effectiveness of spike time-based classification (*Figure 5A*). We trained an MLP on VISp versus VISs, and, in this binarized context, observed a balanced accuracy of 79.98 ±3.03 % (*Figure 5B*). This effect was driven by VISs, which achieved 96% sensitivity. To understand if converting six visual structures into two superstructures truly improved discriminability, it is necessary to make the equivalent comparison in the more complex model. To achieve this, we simply took the results of the visuocortical task across 6 structures (*Figure 4—figure supplement 2*, bottom left) and collapsed the secondary regions into a single class. Intriguingly, this yielded a balanced accuracy of 91.02 ± 0.95%, driven by VISp at 94.4% sensitivity. A likely explanation is that, despite using resampling strategies to correct for class imbalance, VISp initially has more true units than any other structure in the 6-class task, incentivizing accurate classification of it. In contrast, for the binary classification, VISp has fewer units than all secondary structures combined (VISs), which instead incentivises classification of VISs. It seems VISp is a more effective classification target for this problem. These results suggest that the computational differentiability of murine visuocortical regions is at the level of VISp versus secondary areas.

Another major axis along which cortical neurons exhibit anatomical structure is layer. This aligns with recent evidence that indicates that cortical layers can be distinguished computationally, particularly in terms of cell types and tuning preferences (*Schneider et al., 2023*; *Lee et al., 2024*; *Wang et al., 2022*). Thus, we hypothesized that cortical layer might be more reliably embedded in single unit spike times than visuocortical structure. Consistent with this, models trained on visuocortical neurons from all structures were able to robustly recover layer information (*Figure 5C and D*). It is noteworthy that transductive and inductive models achieved similar balanced accuracies on both ISI distribution (inductive unsmoothed balanced accuracy: 46.43 ± 0.97%, inductive smoothed: 62.59 ± 1.13%, transductive unsmoothed: 46.52 ± 0.63%, transductive smoothed: 60.37 ± 2.64%) and concatenated PSTHs (inductive unsmoothed: 41.13 ± 1.28%, inductive smoothed: 52.16 ± 2.38%, transductive unsmoothed: 41.66 ± 0.51%, transductive smoothed: 52.35 ± 1.69%). However, cortical layer information was more robust in the broad ISI distribution, which outperformed concatenated PSTHs, even in transductive models. Also noteworthy is the fact that layer IV exhibited the greatest confusion (specifically, with adjacent layers), achieving a sensitivity of only 28%. These results suggest that general embeddings are more readily available for cortical layers than cortical structures.

## Anatomical embeddings generalize across experimental conditions

Drifting gratings, while widely embraced across decades of research in the visual system, are highly stereotyped and low dimensional compared to natural visual environments. As a result, the anatomical embeddings described thus far could require the repeated presentation of drifting gratings. In other words, it is reasonable to suggest that anatomical information embedded in single-unit activity would be immediately obscured by a complex visual environment. To evaluate this, we trained new, inductive

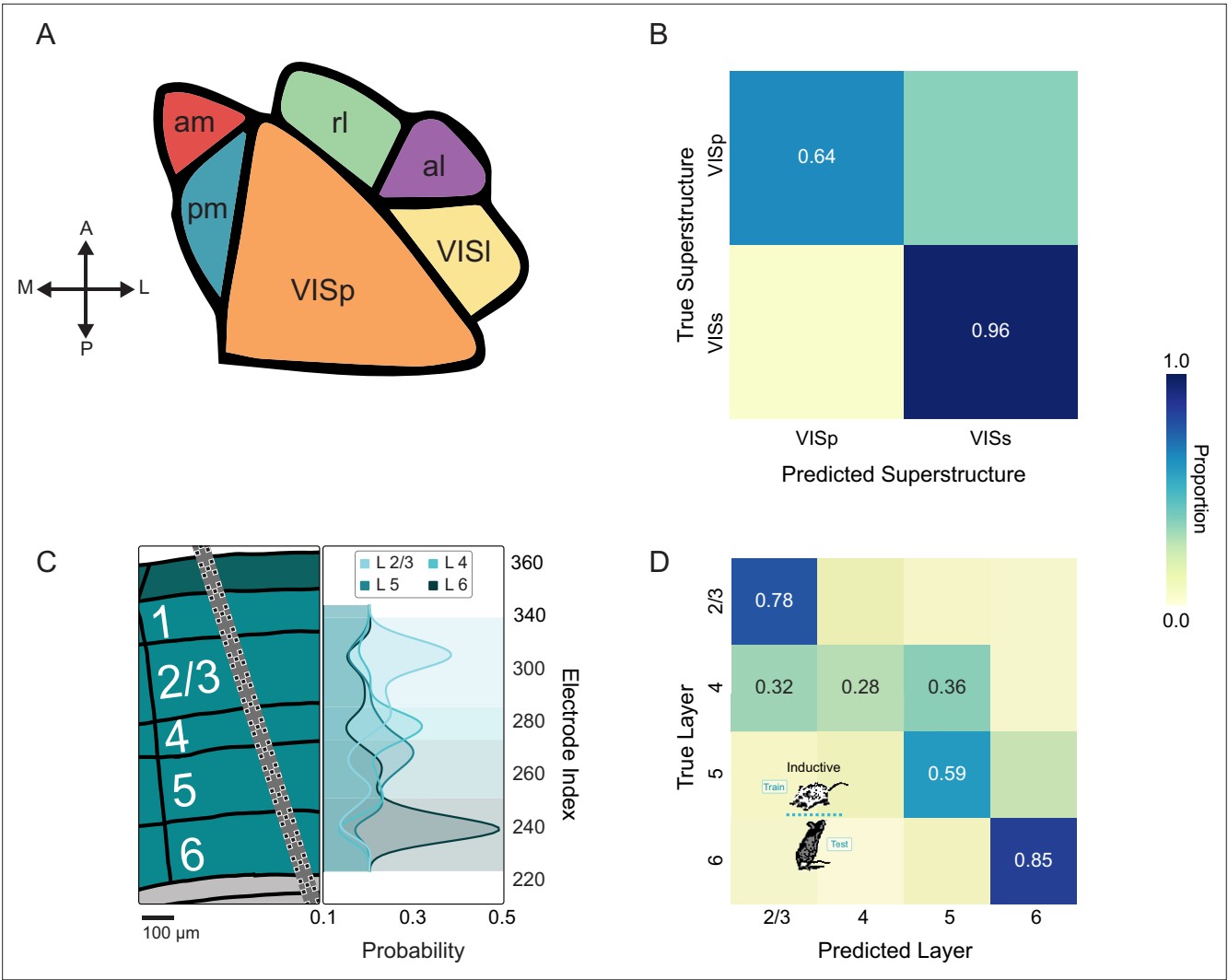

**Figure 5.** Primary vs secondary distinction and cortical layer are more evident in spike timing than individual structures. (**A**) Illustration of visuocortical structures that can be grouped into primary versus secondary superstructures: VIS-am, -pm, -l, -al, -rl are grouped into secondary visual cortex (VISs) while VISp is primary visual cortex. (**B**) Confusion matrix resulting from inductive classification of superstructure (with smoothing). Cells with proportion less than chance (1/number of classes) contain no text. Balanced accuracy is 79.98 ± 3.03%. (**C**) Left: Illustration of example array implanted across cortical layers. Right: Layer probability for the example implant shown on the left calculated by smoothing across neuron-based classifications. Colored background shows the consensus prediction as a function of neuron location (electrode number). (**D**) Confusion matrix resulting from smoothed inductive classification of layer. Balanced accuracy is 62.59 ± 1.12%.

MLPs on single unit activity in the context of drifting gratings as well as two additional conditions: (1) the presentation of naturalistic movies (ten repeated presentations of a 120 s long excerpt from the film *Touch of Evil*), and (2) spontaneous activity (an unchanging gray screen; *Figure 6*). We then tested these MLPs on activity from all three conditions. This allowed us to ascertain (a) whether anatomical information is available outside of the context of drifting gratings and (b) whether the anatomical information learned in one condition can generalize to another.

We tested every pairing of train/test condition on each of six tasks: brain regions, hippocampal structures, thalamic structures, visuocortical structures, visuocortical layers, and visuocortical superstructures. Chance levels varied between 14.3 and 50% across tasks. To facilitate inter-task comparisons, we quantified accuracy by employing Matthews Correlation Coefficient (MCC). MCC is a balanced measure that takes into account true and false positives and negatives, providing a reliable statistical measure especially for imbalanced datasets (*Chicco and Jurman, 2020*; *Boughorbel et al., 2017*). MCC values range from −1 to +1, with +1 representing perfect prediction, 0 indicating performance no better than random chance, and −1 signifying total disagreement between prediction and

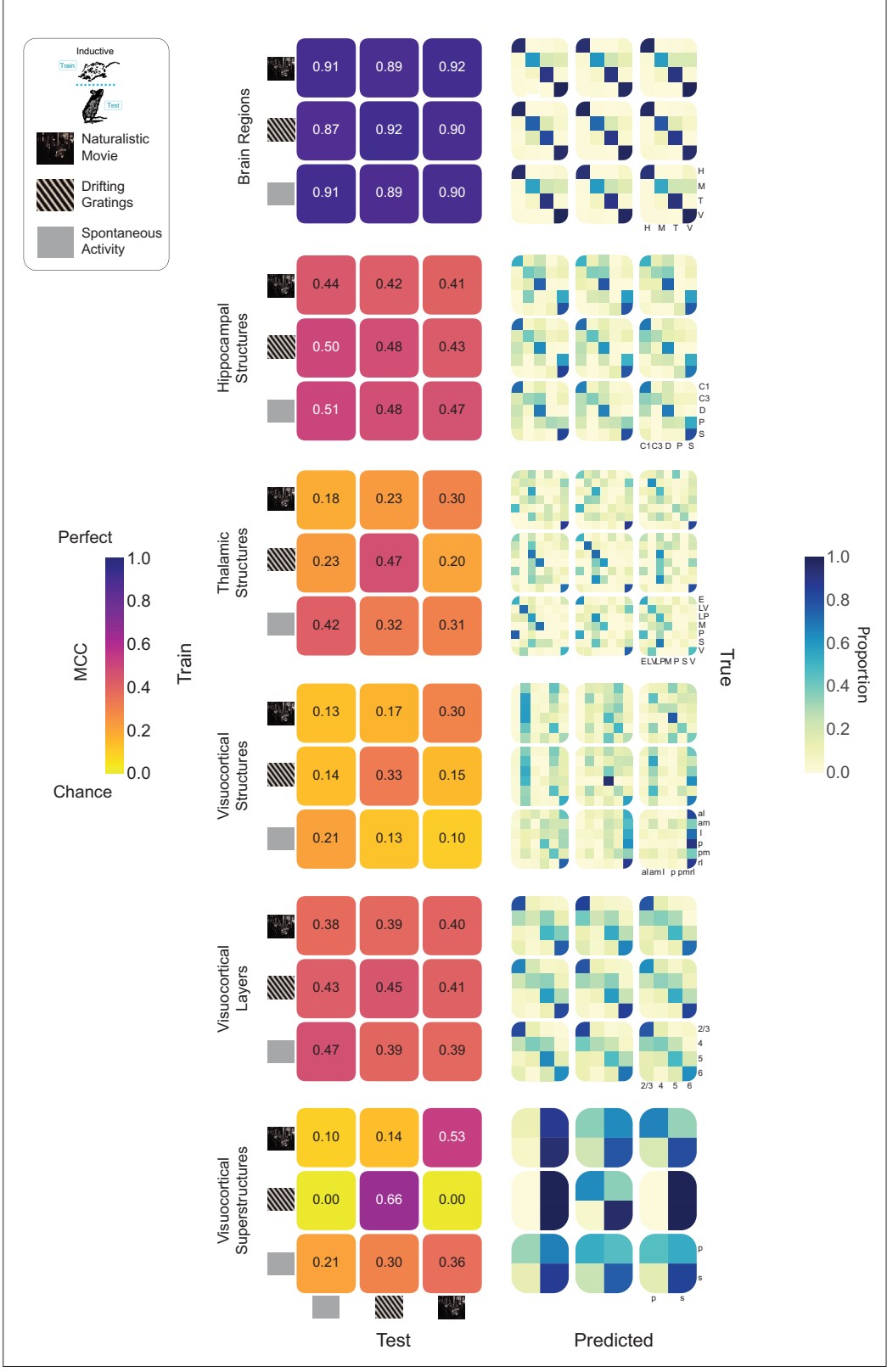

**Figure 6.** Anatomical information embedding in spike trains generalizes across diverse stimuli. (Top Left) Schematic showing inductive train/test split along with visual stimuli: naturalistic movie, drifting gratings, spontaneous activity (i.e. gray screen). (Left Column) Grids showing Matthew's Correlation Coefficient (MCC) values for pairs of training stimuli (grid rows) and testing stimuli (grid columns). Grid diagonals (top right to bottom left)

*Figure 6 continued on next page*

*Figure 6 continued*

represent train/test within the same stimuli. (Right Column) Confusion matrices corresponding to each of the MCC grids in the left column.

The online version of this article includes the following figure supplement(s) for figure 6:

**Figure supplement 1.** The effect of training and testing within and across stimulus conditions.

observation. Importantly, MCC is normalized by the number of categories, allowing for comparisons across tasks with different numbers of classes (*Jurman et al., 2012*).

MLPs across almost every task and every pairing of train/test condition performed far above chance (MCC chance = 0) (*Figure 6*, left column). MLPs tasked with brain region showed remarkable generalizability across stimuli (mean MCC = 0.90). This was followed by hippocampal structures (mean MCC = 0.46), visuocortical layers (mean MCC = 0.41), thalamic structures (mean MCC = 0.30), visuocortical superstructures (mean MCC = 0.26), and visuocortical structures (mean MCC = 0.18). Interestingly, the only instances of chance-level performance arose in the visuocortical superstructure task—MLPs chance (MCC = 0) when trained on drifting gratings and tested on either spontaneous activity or naturalistic movies. While this appears consistent with stimulus-specific embeddings, visuocortical MLPs trained on spontaneous activity were above chance when tested on drifting gratings. The same was true for those trained on naturalistic movies. Taken together, these data suggest that the embedding of anatomical information in single neuron activity is not abolished by complex visual stimuli or spontaneous activity, and that the embeddings learned in one context are not absent in other contexts. The visuocortical superstructure task results imply that, in some contexts, complex stimuli may produce more generalizable results than simple stimuli.

In each of the 54 combinations of task and train/test condition, there were diverse underlying patterns of MLP learning (*Figure 6*, right column). Intuitively, instances with high MCC, such as those in the brain regions task, produced strong diagonal structure (where the predicted class aligns with the true class). Tasks resulting in lower MCC, such as the visuocortical structures, yielded slight diagonal structure in some cases, but were driven by a small number of accurate points. Across all six tasks, within-stimulus models (e.g. trained on spontaneous, tested on spontaneous) tended to produce significantly greater MCCs than across-stimulus models, although all were above chance (*Figure 6— figure supplement 1*).

## Anatomical embeddings generalize across research laboratories

The results of inductive models trained and tested in mismatched conditions suggest that, amidst stimulus information, neuronal spike trains carry universal signatures of their anatomical location. If this were true, it should be possible to apply models trained on animals from one research group—in this case, the Allen Brain Observatory team—and decode neuronal anatomy from data generated at an independent location. In other words, the Allen-based model should predict a neuron's anatomical location based on its spike train even if the recording was conducted in a different location and experimental context.

To test this, we required independently generated data that maintained two key features: (1) the spatiotemporal micro-scale resolution of the Allen Institute's recordings, and (2) the anatomical breadth of the Allen Institute's recordings. These criteria were satisfied by an open dataset from *Steinmetz et al., 2019*, which comprises high-density silicon recordings that span many of the same regions and structures examined in the Allen data. However, the experiments (*Steinmetz et al., 2019*) are markedly different, in that they comprise mice trained to carry out a decision-making task. Summarily, the task involved observing a Gabor patch of fixed orientation (45°) and spatial frequency (0.1 cycles per degree) with varying contrast on either the left or right side, in response to which the mouse must turn a wheel toward the side with higher contrast (*Steinmetz et al., 2019*). Recall that the Allen data were recorded in the context of passive viewing (*Figure 7A*).

We classified anatomy in *Steinmetz et al., 2019* data using two variants of our Allen-based models; one trained on ISI distributions recorded during the presentation of drifting gratings, and one trained on ISI distributions recorded in three conditions (drifting gratings, naturalistic movies, and spontaneous activity). Both models successfully predicted brain region above chance in every Steinmetz et al. animal (N=10), with the exception of one animal in the drifting gratings model (mean

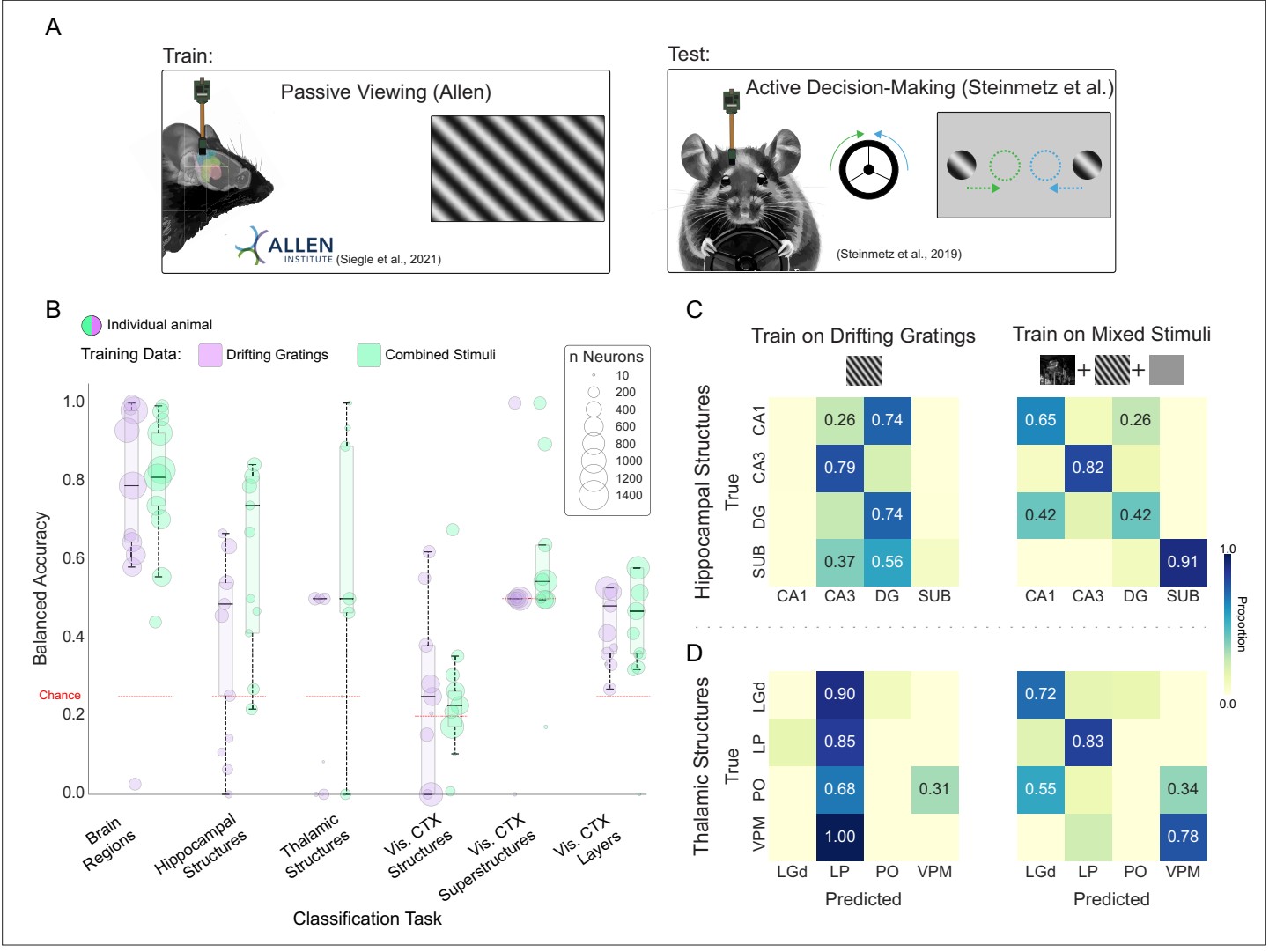

**Figure 7.** Anatomical information embedded in spike trains generalizes across laboratories and protocols. (A) Illustration of behavioral tasks employed by two laboratories: (left) Passive Viewing (Allen Institute) vs. (right) Active Decision-Making (Steinmetz et al.). In Active Decision-Making, the mouse spins a wheel in response to the location of the drifting grating presented on the screen. (B) Bubble chart showing balanced accuracy of individual test set animals from the Steinmetz et al. The model was trained on Allen Institute data. Bubble size indicates the number of units recorded in the test animal. Black horizontal lines represent the median of the balanced accuracy distributions across the animals. Models are trained with either drifting gratings alone (purple) or a combination of drifting gratings, natural movies, and spontaneous/gray screen (green). Balanced accuracy across all Steinmetz et al. neurons: BR DG = 80.46%, BR Mix: 81.28 %, HS DG: 40.07 %, HS Mix: 69.89 %, TS DG: 21.52 %, TS Mix: 58.22 %, VCS DG: 28.41 %, VCS Mix: 28.34 %, VCSS DG: 58 %, VCSS Mix: 59 %, VC Layers DG: 46.36 %, VC Layers Mix: 49.01% where BR stands for brain regions, HS hippocampal structures, TS thalamic structures, VCS visual cortex structures, VCSS visual cortex superstructures, DG drifting gratings and Mix denotes the combined stimuli. (C) Confusion matrices for prediction of hippocampal structures in Steinmetz et al. test set when trained on drifting gratings stimulus (left) vs. mixed stimuli (right) from the Allen Institute. (D) Confusion matrices for prediction of thalamic structures in Steinmetz et al. test set when trained on drifting gratings stimulus (left) vs. mixed stimuli (right) from the Allen Institute.

The online version of this article includes the following figure supplement(s) for figure 7:

**Figure supplement 1.** Train/test across laboratories: Allen-to-Steinmetz et al. confusion matrices.

balanced accuracy calculated across neurons, drifting gratings model—80.46%; combined stimuli—81.28%; *Figure 7B*). At the level of structures, two general principles emerged: (1) models trained on combined stimuli were generally more effective than drifting gratings models— this is particularly evident when examining diagonal structure in the confusion matrices (*Figure 7B and C*, 6.8), and (2) the visuocortical structures task did not transfer effectively between laboratories (drifting gratings model—28.41%, mixed model—28.34%). Hippocampal and thalamic structures as well as

visuocortical layers were identifiable well above chance, especially in mixed models (drifting gratings/mixed models: hippocampal structures—40.07/69.89%; thalamic structures—21.52/58.22%; visuocortical layers—46.36/49.01%). Visuocortical superstructures were marginally identified, but only in the combined stimuli model (drifting gratings model—58.03%, mixed model—59.08%), as only one animal was above chance in the drifting gratings condition (*Figure 7B*). Note that, due to differences in the recordings and number of units from the structures, each Steinmetz et al. task involved four classes, such that chance is 25% with the exception of visuocortical structures ($k$=5, chance = 20%) and visuocortical superstructures ($k$=2, chance = 50%). Finally, error is not reported as there are no splits to test across in the Steinmetz et al. data; balanced accuracy is reported for the entire applicable dataset.

Taken together, these data suggest that information about a neuron's anatomical location can be extracted from the time series of its spiking. This principle appears to apply to neurons from diverse structures across the telencephalon, diencephalon, and mesencephalon. Further, some of the rules by which this information is organized are shared across animals and are not an artifact of a task or local protocol.

## Discussion

Understanding the information carried within a neuron's spiking has been the subject of investigation for a century (*Adrian and Bronk, 1928*). While it is well established that neuronal activity encodes stimuli, behavior, and cognitive processes (*Faisal et al., 2008*; *Gerstner et al., 2014*), the possibility that spike trains might also carry information about a neuron's own anatomical identity has remained largely unexplored. Our study provides compelling evidence that such anatomical information is indeed embedded within neuronal spike patterns. Using machine learning approaches, we demonstrate that this embedding is robust across multiple spatial scales, from broad brain regions to specific structures and cortical layers. Crucially, these anatomical signatures generalize across animals, experimental conditions, and even research laboratories, suggesting a fundamental principle of neural organization. Our findings reveal a previously unrecognized dimension of the neural code, one that is multiplexed with the encoding of external stimuli and internal states (*Olshausen and Field, 2004*; *Harris and Thiele, 2011*; *Parks et al., 2023*). These data advance an understanding of the relationship between structure and function in neural circuits, raising the possibility that structure is not ancillary to neuronal computation, but intrinsically embedded within it. Beyond fundamental scientific insight, our findings may be of benefit in various practical applications, such as the continued development of brain-machine interfaces and neuroprosthetics, as well as for the interpretation of large-scale neuronal recordings. Note, however, that a purely utilitarian goal, for example electrode localization in electrophysiological recordings, would be well served by considering additional features, such as extracellular waveforms (*Buccino et al., 2018*; *Jia et al., 2019*), in addition to spike timing. While this approach holds promise for practical application, the inclusion of waveform information subverts the question of whether a neuron's output—the timing of its spiking–contains an embedding of its location.

Clearly, there are powerful differences throughout the brain as a function of anatomy. There is extensive literature to describe this diversity at many levels, from gene expression gradients in hippocampal pyramidal neurons (*Cembrowski et al., 2016*) to distinct connectivity patterns across cortical layers (*Harris et al., 2019*). A complementary albeit smaller literature provides some indication that, at a coarse-grained level, similar gradients can be identified in neuronal activity. For example, examined at the level of the entire population, there are subtle but reliable differences in isocortical neuronal variability in functionally distinct cortical areas (*Shinomoto et al., 2009*). Similarly, considered as a population-level statistic, there is a cortical gradient of neuronal intrinsic timescale (a measure of how long a neuron's activity remains correlated with itself over time) (*Murray et al., 2014*). However, while the ability to detect statistical differences across a population demonstrates that anatomy bears some influence on brain activity, it does not suggest that anatomy can be extracted from a single neuron's activity. This is exemplified by placing two observations side by side. First, single neuron firing rates are log-normally distributed. This means that within any given brain region, there is a wide range of firing rates, with a long tail of high-firing neurons. Second, the population-level distribution of firing rates varies slightly but significantly as a function of region (*Roxin et al., 2011*; *Mizuseki and Buzsáki, 2013*). The combination of these facts implies that while there may be detectable differences between

regions at the population level, the extensive overlap in firing rate distributions makes it challenging, if not impossible, to determine a neuron's anatomical origin based solely on its firing rate.

Here, we approach this problem from a fundamentally different perspective than the population-based paradigm. Capitalizing on (1) the availability of open, high-density recordings across a plurality of brain regions and (2) machine learning methods capable of learning complex, nonlinear relationships, we approach the problem at the level of single neuron classification. This approach has strengths and weaknesses. The principal strength is the revelation that anatomical information is embedded in the activity of individual neurons throughout the brain. The principal weakness is limited interpretability, a criticism widely raised when contemplating machine learning (even when addressing this issue, the ground-truth patterns can themselves be complex and non-intuitive, e.g., *Figure 3*). However, in exchange for simplicity, our approach is founded on establishing whether there exist spike train-based anatomical signatures that generalize across animals, experimental conditions, and research laboratories. That such signatures are readily identifiable indicates neurons are not mere conduits of stimulus-related information. While neurons can be described as rate-varying Poisson processes (*Softky and Koch, 1993*), it is increasingly clear that a more complete description should incorporate additional streams of information, such as transcriptomic cell type and anatomy across varying time scales (*Schneider et al., 2023*; *Mi et al., 2023*).

Our findings reveal a striking difference in the generalizability of anatomical information encoded in ISI distributions versus PSTHs. Specifically, inductive models trained on ISI distributions maintained performance levels comparable to their transductive counterparts across all tasks, while PSTH-based models showed a significant drop in performance when tested on new animals. This is perhaps surprising, given that stimulus response properties (such as receptive field size and response latency) are generally understood to exhibit some consistency across animals recorded under equivalent conditions (*Niell and Stryker, 2008*; *Schmolesky et al., 1998*; *Siegle et al., 2021*). Viewed through the lens of stimulus and response, PSTHs in the Allen Institute datasets are robust to slight differences in retinotopic locations across animals (*Siegle et al., 2021*). Despite this, our PSTH-based models performed poorly when trying to predict anatomical location in new animals under the same stimulus conditions. Thus, it is likely that stimulus-response information is dissociable from the embedding of anatomical location.

The difference in generalizability when comparing PSTH- and ISI-based models may reflect a fundamental distinction in the nature of information captured by these two representations of neural activity. ISI distributions encapsulate intrinsic properties of neuronal firing patterns that appear to be conserved across animals, potentially reflecting stable, anatomy-specific computational features. These might include cell types and their resultant ion channel compositions (*Zeisel et al., 2018*; *Saunders et al., 2018*), local circuit motifs (*Braganza and Beck, 2018*; *Luo, 2021*), or homeostatic mechanisms that shape firing statistics independent of stimuli (*Hengen et al., 2013*; *Hengen et al., 2016*; *Ma et al., 2019*). Crucially, these features are not inherently tied to a stimulus. In contrast, PSTHs involve studying the repeated presentation of the same stimulus. Despite this repetition, there is neuron and trial level variability (*Renart and Machens, 2014*; *Cohen and Kohn, 2011*). Neuronal response variability to repeated stimuli often correlates among nearby neurons (*Smith and Kohn, 2008*; *Smith and Sommer, 2013*) and is influenced by local circuit connectivity and the animal's cognitive factors such as attention (*Rosenbaum et al., 2017*; *Ruff and Cohen, 2014*). Thus, transductive models may learn from variability in the training dataset that is highly informative in the test dataset. However, such variability would not carry across animals. The robustness of ISI-based anatomical embeddings across animals and even across laboratories underscores the fundamental nature of these anatomical fingerprints in neural activity, transcending individual differences and specific experimental paradigms.

Our analysis of visual isocortical structures offers intriguing insights into the computational organization of the murine visual system. While PSTH-based models performed well in classifying visual areas, suggesting stimulus-specific differences in neuronal responses across these regions, the overall classification accuracy remained relatively low compared to other brain areas. This finding aligns with previous studies that have demonstrated differences in orientation and spatial frequency tuning across mouse visual areas (*Marshel et al., 2011*; *Andermann et al., 2011*; *Roth et al., 2012*; *Ayzenshtat et al., 2016*). However, our results suggest that these differences, while statistically significant at the population level, may not translate into robust, neuron-specific computational signatures. The murky distinction between primary and secondary visual areas in mice is reflected in our data, with the most

reliable discrimination occurring between primary visual cortex (VISp) and grouped secondary areas, rather than among individual secondary regions. Even this distinction, however, was not dramatic. Interestingly, we found that isocortical layers within visual areas were more readily distinguishable than the areas themselves, suggesting that laminar organization may play a more fundamental role in shaping neuronal computations than areal boundaries in mouse visual cortex. This hierarchical organization - from broad regions to superstructures (e.g. VISp vs. secondary areas) to substructures (layers) - provides a nuanced view of functional specialization in the mouse visual system. While numerous recent studies have indicated functional specialization of mouse higher visual areas beyond VISp (*Wang and Burkhalter, 2007*; *Glickfeld and Olsen, 2017*; *Kumar et al., 2021*), our data suggest that these specializations may not manifest as significant differences in single neuron spiking features across areas. This observation raises the possibility that secondary visual areas in mice may not be computationally distinct, at least in terms of their constituent neurons' fundamental spiking patterns. The relative inability to classify the structure of visual cortical neurons may, in fact, reflect a genuine neurobiological feature: the computational properties of neurons in different visual areas may be largely indistinguishable based on spike timing alone.

Our findings demonstrate that individual neurons embed information about their anatomical location in their spike patterns, with these anatomical embeddings conserved across animals and experimental conditions. This previously unrecognized aspect of neural coding raises questions about the relationship between brain structure and function at the single-neuron level. Future research will be crucial in determining whether and how different streams of information, including these anatomical embeddings, contribute to neural computation and the variegated functions of the brain. While the technical barriers are daunting, the obvious experiment is to manipulate a neuron's anatomical embedding while minimally impacting external and internal variables, such as stimulus information and levels of neurotransmitters or neuromodulators. Should this disrupt any aspect of sensation, perception, cognition, or behavior, the answer would be clear. If not, there is still great practical utility in an experimenter's capacity to ascertain anatomy based on neuronal activity.

## Methods

### Datasets

Neurodata Without Borders (NWB) files for the Allen Institute's Visual Coding Neuropixels dataset were retrieved with AllenSDK (*Allen Institute for Brain Science, 2019*). Units passing the default filtering criteria of ISI violations < 0.5, amplitude cutoff < 0.1, and presence ratio > 0.9 were selected for further analysis. Units with a firing rate below 0.1 Hz over the session were excluded.

Mice were head fixed and presented with visual stimuli for a roughly 3-hr session (*Siegle et al., 2021*). The mice were shown one of the two slightly different visual stimulus sets: 'Brain Observatory 1.1' (N=32 animals) or 'Functional Connectivity' (N=28 animals; *Allen Institute MindScope Program, 2019a*; *Allen Institute MindScope Program, 2019b*). From these sessions, we retrieved spike times coincident with the presentation of drifting gratings and natural movie three from 'Brain Observatory 1.1' and gray screen (spontaneous) from both sets. For each animal, 30 min of drifting gratings were presented. Each trial consisted of 1 s of gray screen followed by 2 s of drifting gratings. Gratings were varied in spatial orientation (0°, 45°, 90°, 135°, 180°, 225°, 270°, 315°) and temporal frequency (1, 2, 4, 8, and 15 Hz), but consistent in spatial frequency (0.04 cycles/degree) and contrast (80%) with 15 equivalent presentations of each particular stimulus. Separately, a sustained mean-luminance blank (gray) screen was presented during intervals between blocks of stimuli. In total, the gray screen was presented in this manner for approximately 20 min to each animal. Natural movie three consisted of a 120 s clip from the opening scene of the movie Touch of Evil (*Welles, 1958*) repeated 10 times. After passing through the criteria above, there were 18,961 units for drifting gratings and natural movie three (visual cortex = 9402, hippocampus = 4301, thalamus = 4068, and midbrain = 920) while the spontaneous stimulus contains 37,458 units (visual cortex = 18,514, hippocampus = 10,337, thalamus = 6625, and midbrain = 1982). In order to have at least 30 units in the test for each split, we removed classes (regions or structures) from the dataset if they contained less than 150 units. We also removed units with ambiguous structure assignment such as 'VIS' or 'TH'. The final number of units in each structure (for drifting gratings and natural movie three) is as summarized in *Table 1*. The spontaneous

**Table 1.** Number of included single units as a function of brain structure: Allen Institute data.

| Region | Structure | Number of Units |
|---|---|---|
| Hippocampus | CA1 | 2,552 |
| | CA3 | 367 |
| | DG | 713 |
| | ProS | 176 |
| | SUB | 470 |
| Midbrain | MB | 920 |
| Thalamus | Eth | 225 |
| | LGd | 903 |
| | LP | 1462 |
| | MGv | 175 |
| | PO | 289 |
| | SGN | 178 |
| | VPM | 261 |
| Visual Cortex | VISal | 1595 |
| | VISam | 1537 |
| | VISl | 971 |
| | VISp | 2076 |
| | VISpm | 904 |
| | VISrl | 1467 |

(blank screen) stimulus has more units from the 'Functional Connectivity' dataset in addition to the units from the 'Brain Observatory'.

The dataset provides labels for each unit's brain structure as ecephys_structure_acronym, determined through a combination of stereotactic targeting, histology, common coordinate framework (CCF) mapping, and intrinsic signal imaging (particularly for visual cortical areas; *Siegle et al., 2021*; *Allen Institute MindScope Program, 2019b*). For our larger brain region prediction tasks, we grouped the provided brain structures into a higher hierarchy based on the Allen Brain Atlas ontology's structure tree (*Wang et al., 2020*). However, the publicly available units' metadata does not include layer information. To address this for cortical units, we mapped the CCF coordinates of each unit onto a finer-grained parcellation (25 µm) level within the ontology, enabling the extraction of layer labels. There were 1229 units, 1601 units, 3046 units, and 1057 units in layers 2, 4, 5, and 6, respectively, for drifting gratings and natural movie three.

To generalize across experimental conditions, we used a dataset from *Steinmetz et al., 2019*, which generally inspired the experiments used in the International Brain Laboratory (IBL) datasets. The mice were shown drifting gratings (oriented at 45° and spatial frequency of 0.1 cycles per degree) of varying contrast. The tasks were divided into an average of four second trials and involved presentation of auditory cue, wheel turning, and reward presentation with inter-trial intervals of gray screen (*Steinmetz et al., 2019*). The experimental sessions also involved passive stimulus presentation after the behavioral sessions (*Steinmetz et al., 2019*). We did not restrict our analysis to any interval but used all the available spikes in the experiments. The data is collected from 10 mice and 39 sessions, with each session potentially resulting in different units from the same mice. The units (or clusters) have structure labels or anatomical locations which were determined by combining electrophysiological features, histology, and CCF mapping (*Steinmetz et al., 2019*). To extract layer information, we used the same method as with the Allen Institute's dataset: mapping the CCF coordinates to higher resolution parcellation using the Allen CCF reference space. We observed that some brain structure labels provided in the dataset did not match the coordinate mappings, possibly due to manual adjustments.

**Table 2.** Number of included single units as a function of brain structure: Steinmetz et al. data.

| Region | Structure | Number of Units |
|---|---|---|
| Hippocampus | CA1 | 1078 |
| | CA3 | 511 |
| | DG | 548 |
| | SUB | 541 |
| Midbrain | MB | 108 |
| Thalamus | LGd | 134 |
| | LP | 395 |
| | PO | 619 |
| | VPM | 68 |
| Visual Cortex | VISam | 1358 |
| | VISl | 535 |
| | VISp | 1535 |
| | VISpm | 540 |
| | VISrl | 249 |

Since accurate layer extraction required the provided structures to overlap with the CCF-mapped structures, we decided to use only those units where the labels matched. In addition, we removed the structures that were not in the training set, that is structures not found in our final Allen dataset. This resulted in 8219 units with the number of units in each structure as summarized in *Table 2*.

## Exploratory visualization

Dimensionality reductions across spiking features utilized python implementations of principal component analysis (PCA; *Pedregosa et al., 2011*) and linear discriminability analysis (LDA; *Fisher, 1936*). Features to be dimensionally reduced were either a collection of established spiking metrics (*Schneider et al., 2023*; see Logistic Regression section for more details), ISI distributions (100 uniform bins between 0 and 3 s; see Multi-Layer Perceptron section), and averaged PSTHs (100 uniform bins over the 3 s span of a trial; see Multi-Layer Perceptron section) for each stimulus condition grouped/concatenated together.

For scatterplots, after computation of the values in the spaces, these values were capped between the 1st and 99th percentiles of the data solely for visualization. This was done because outliers cause the central tendency of the data and its separability by region/structure to be less visually apparent. Points affected by this cap remain visible at the bounds of these axes. This was done solely for visualization purposes and did not affect the construction of the dimensionally reduced spaces.

To evaluate whether the ground truth labels of structures or regions emerged as clusters in the dimensionally reduced spaces. To do this, we performed k-means clustering in the dimensionally reduced space with k equal to the number of ground truth classes. We then calculated the adjusted r and index between the region/structure labels and the cluster labels. We also sought to evaluate whether the relationship (relative distances) between regions/structures was preserved between classes across feature sets and dimensionality reduction methods. First, we created histograms of the distribution of scores along the primary axis of the dimensionality reduction for each region/structure. For each pair of regions/structures, we calculated the Wasserstein distance between their histograms. We plotted the results as triangular distance matrices to show that these relationships are largely unconserved across feature sets and dimensionality reduction methods.

## Dataset splitting

To properly optimize our supervised models while ensuring generalizability to unseen data, we performed a standard split into train, validation, and test sets (with an approximate 60/20/20 ratio

with respect to the number of neurons). To ensure the full dataset was evaluated, we extracted five stratified samples such that each unit was included in the validation set in one of these splits and separately included in the test set in another split. This is intuitively similar to a fivefold cross-validation.

In a transductive split, for each animal, 60% of the neurons were allocated to train, 20% of the neurons were in the validation set, and 20% were allocated to the test set. We sought to ensure classes (i.e. regions/structures) were represented appropriately in each set. If 20% of the dataset was class A, we attempted to stratify our sets such that ~20% of the train set was class A, ~20% of the validation set was class A, and ~20% of the test set was class A.

In an inductive split, all neurons from 60% of the animals were allocated to train, all neurons from 20% of the animals were allocated to validation, and all neurons from 20% of the animals were allocated to the test set.

For the generalization across experimental conditions, 80% of the Allen units were allocated to a training set, while 20% were used as a validation set. All of the Steinmetz et al. units in the final dataset were used as the test set.

## Logistic regression

Summary statistics based on spike times during drifting gratings presentation were obtained and used as features for a logistic regression implemented in scikit-learn (*Pedregosa et al., 2011*). This workflow largely follows our prior work (*Schneider et al., 2023*). The logistic regression used L2-regularization and 1E-4 tolerance stopping criterion. When possible, interspike intervals were used to compute these statistics instead of binned spike counts. The following standard statistics were computed: mean firing rate, standard deviation of ISI, median ISI, maximum ISI, minimum ISI, coefficient of variation. Oscillatory activity of the spike train was captured using the power spectral density (PSD) from a periodogram using scipy.signal (*Virtanen et al., 2020*). The mean value was calculated within each of the following bands (< 4 Hz, 4–8 Hz, 8–12 Hz, 12–40 Hz, 40–100 Hz), and the ratio of each band's power to the power across all bands < 100 Hz was used as a feature. Local variability in the spike train was captured through the CV2, LV, and LVR statistics (as implemented in Elephant; *Denker et al., 2018*). These statistics were used first individually, and second aggregated to predict each class (structure/region) for a task.

## Multi-layer perceptron

For each task, an MLP was implemented in scikit-learn (*Pedregosa et al., 2011*) with a designated set of features to represent the spike train. Features were either: (1) interspike intervals (ISIs) distribution, (2) averaged peri-stimulus histogram (PSTH), or (3) concatenated PSTHs.

To calculate ISI distribution for a neuron under specific stimulus, we converted all the spike times of that neuron during the stimulus's presentation into ISIs. Then, we created 100 uniform bins between 1ms and 3 s and counted the neuron's ISIs in each bin. This resulted in 300 features of ISI 'distribution' for each neuron. Finally, we performed min-max normalization, where for each neuron, the maximum value across all bins was set to 1, and the minimum value was set to 0.

The average and concatenated PSTHs were calculated for the drifting grating stimulus. For the average PSTH, we aligned all spikes around the stimulus presentation (1 s before the stimulus presentation and 2 s of stimulus presentation) for each neuron. Then, we created 100 uniform bins over 3 s and counted the number of spikes in each bin for each trial. Then, we averaged the values in each bin across all trials. This resulted in 300 features for each neuron. Finally, we performed min-max normalization, where the maximum value across all bins was set to 1, and the minimum value was set to 0. For the concatenated PSTH, a PSTH was calculated separately for each combination of the drifting grating's parameters, that is temporal frequency and orientation. Similar to the average PSTH, we aligned the spikes around the stimulus presentation and performed a binned spike count for 30 uniform bins between 0 and 3 s. We then constructed the PSTH by taking the mean across the number of trials. As each combination of the drifting grating's parameter was repeated 15 times during the experiment, the PSTHs are the mean of 15 trials. Similar to the ISI distribution and average PSTH, we performed min-max normalization for each neuron. This operation took place on each PSTH prior to concatenation. Given that there were 40 parameter combinations and 30 bins for each combination, this resulted in 1,200 features per neuron.

**Table 3.** MLP hyperparameters.

| Hyperparameter | ISI Distribution | Avg. PSTH | Cat. PSTH |
|---|---|---|---|
| Number of Nodes | 50:600:50 | 30:300:50 | 50:600:50 |
| Learning Rate | 1E-7,1E-6,1E-5,1E-4,1E-3,1E-2,1E-1 | 1E-5,1E-4,1E-3,1E-2,1E-1 | 1E-5,1E-4,1E-3,1E-2,1E-1 |
| Batch Size | 25:500:50 | 50:600:50 | 50:600:50 |
| Alpha | 1E-4,1E-3,1E-2,1E-1,1E0,5,10 | 1E-4,1E-3,1E-2,1E-1,1E0 | 1E-4,1E-3,1E-2,1E-1,1E0 |
| Beta_1 | 0.5:0.9:0.1 | 0.5:0.9:0.1 | 0.5:0.9:0.1 |

For each task, a Bayesian hyperparameter optimization (HyperOpt package) (*Bergstra et al., 2013*) was used to tune hyperparameters (*Table 3*) such that they maximized balanced accuracy on the validation set. To mitigate the effect of class imbalance in our dataset, we treated resampling as a hyperparameter and evaluated undersampling, oversampling, and data augmentation. We found that the synthetic minority oversampling technique (SMOTE; *Chawla et al., 2002*), a data augmentation approach, yielded optimal performance on validation sets. Therefore, classes were resampled with SMOTE in all cases unless otherwise specified. A different model was trained and tuned for each of five splits to avoid data leakages. For our inductive predictions across datasets (Allen to Steinmetz et al.) tasks, to ensure that the training data encompassed the relevant anatomical representations present in the Steinmetz et al. dataset, we trained models on a subset of the Allen dataset that included only brain structures that are also present in the Steinmetz et al. dataset.

Consistent with prior work (*Schneider et al., 2023*), an MLP with a single hidden layer was generally found to perform better than an MLP with multiple hidden layers. The number of nodes for this hidden layer was optimized. Alpha is the L2 regularization term. Beta_1 is the exponential decay rate for estimates of the first moment vector in the Adam optimizer. All unmentioned hyperparameters took on default values from sklearn.

To identify the features contributing to the models' learning, we employed permutation feature importance, which measures the extent to which shuffling a specific feature across samples increases the model's prediction error (*Fisher et al., 2018*). To quantify the feature importance of a specific inter-spike interval (ISI) distribution bin for predicting a particular brain region or structure, we first estimated the model's error for the original features, given that the neurons originated from that specific brain region or structure. Then, for those neurons, we randomly permuted (shuffled) a specific bin of the ISI distribution across all samples (neurons) and estimated the model's error for the permuted features. The difference between the original and the permuted features' error represents the feature importance for that specific bin and region/structure. We repeated this process 100 times for all bins in the ISI distribution, each region/structure, and across the five splits. The feature importance is the mean across the splits and repetitions. We followed a similar approach for the concatenated PSTH.

## Smoothing

For some tasks, predictions were spatially smoothed to improve predictions by leveraging the spatial arrangements of recording channels to weight the influence of nearby units on each prediction. First, neurons (units) were grouped by session and probe to ensure the smoothing was applied within the same probe and session. Second, the predicted probabilities for each class were averaged for all neurons sorted to a particular electrode. Electrodes with no neurons detected were ignored for the purposes of this smoothing. Third, for each electrode, the spatial proximity of nearby units was leveraged by calculating a Gaussian weight. For an electrode, the weight for each nearby unit was determined using a normal distribution centered on the current unit's electrode index (i.e. a linear approximation of the electrode geometry) with a standard deviation equal to the smoothing window. The probability density function (PDF) of the normal distribution was used to compute these weights. The class probabilities for each nearby unit were multiplied by their respective weights. The weighted probabilities were then summed and normalized by the sum of the weights to produce the smoothed prediction for each class for that electrode. The process was repeated for all electrodes. To determine an optimal standard deviation of the Gaussian kernel, we used Hyperopt to maximize balanced accuracy on the validation set.

## Models performance metrics and statistical tests

In order to quantify the models' performances, we mainly used balanced accuracy, which is the macro average of recall (true positive rate) per class (*Brodersen et al., 2010*). Chance level is 1/number of classes. In some cases, we used the Matthews Correlation Coefficient (MCC). This measure considers true positives (TP), true negatives (TN), false positives (FP), and false negatives (FN). It provides a balanced measure, useful even with imbalanced datasets, with a value ranging from –1 (total disagreement) to 1 (perfect prediction) (*Chicco and Jurman, 2020*; *Boughorbel et al., 2017*). 0 represents chance-level (which is adjusted to the number of classes). As most of our tasks were multiclass classification, we used a multi-class version of MCC (*Pedregosa et al., 2011*; *Gorodkin, 2004*).

Given a confusion matrix $C$ for $N$ classes, the following intermediate variables are defined:

$$t_n = \sum_{i=1}^{N} C_{in} \quad \text{(total occurrences of class } n \text{ in the actual data)},$$

$$p_n = \sum_{i=1}^{N} C_{ni} \quad \text{(total occurrences of class } n \text{ in the predictions)},$$

$$c = \sum_{n=1}^{N} C_{nn} \quad \text{(total number of correctly predicted samples)},$$

$$s = \sum_{i=1}^{N} \sum_{j=1}^{N} C_{ij} \quad \text{(total number of samples)}.$$

The multi-class MCC is defined as:

$$\text{MCC} = \frac{c \times s - \sum_{n=1}^{N} p_n \times t_n}{\sqrt{\left(s^2 - \sum_{n=1}^{N} p_n^2\right) \times \left(s^2 - \sum_{n=1}^{N} t_n^2\right)}}.$$

In all cases, the statistical test and level of significance are indicated in the relevant sections of the main text and figure legends. In most cases, linear mixed effects models are employed with subsequent ANOVA for main effect and post-hoc EMMeans with Tukey test for pairwise comparisons. The implementation was in R (lmer; *Bates et al., 2015*). In some cases, multiple T-tests with Bonferroni correction (as appropriate) were employed. The implementation was in Python (scipy.stats; *Virtanen et al., 2020*).

## Acknowledgements

We would like to thank the Allen Institute for generating and sharing the Visual Coding datasets. We would like to thank Dr. Josh Siegle for technical insights into the Allen datasets and his helpful perspective on our work. We would also like to thank Dr. Nick Steinmetz for sharing data and technical advice. We are grateful for the funding that enabled this work: R01NS118442 (KBH), R01EB029852 (ELD & KBH), F31NS134240 (AMS), the Cognitive, Computational and Systems Neuroscience (CCSN) Fellowship (GBT), and the Incubator for Transdisciplinary Futures, an Arts & Sciences signature initiative at Washington University in St. Louis (KBH).

## Additional information

### Funding

| Funder | Grant reference number | Author |
|---|---|---|
| BRAIN Initiative | R01NS118442 | Keith B Hengen |
| BRAIN Initiative | R01EB029852 | Eva Dyer<br>Keith B Hengen |

| Funder | Grant reference number | Author |
|---|---|---|
| National Institutes of Health | F31NS134240 | Aidan M Schneider |
| Washington University in St Louis | CCSN Program | Gemechu Bekele Tolossa |
| Washington University in St Louis | Incubator for Transdisciplinary Futures | Keith B Hengen |

The funders had no role in study design, data collection and interpretation, or the decision to submit the work for publication.

## Author contributions

Gemechu Bekele Tolossa, Aidan M Schneider, Conceptualization, Data curation, Formal analysis, Investigation, Visualization, Methodology, Writing – original draft, Writing – review and editing; Eva Dyer, Conceptualization, Supervision; Keith B Hengen, Conceptualization, Supervision, Funding acquisition, Visualization, Writing – original draft, Project administration, Writing – review and editing

## Author ORCIDs

Gemechu Bekele Tolossa ⓘ https://orcid.org/0000-0001-6405-2908
Keith B Hengen ⓘ https://orcid.org/0000-0001-5017-4090

Reviewer #1 (Public review): https://doi.org/10.7554/eLife.101506.3.sa1
Reviewer #2 (Public review): https://doi.org/10.7554/eLife.101506.3.sa2
Author response https://doi.org/10.7554/eLife.101506.3.sa3

# Additional files

## Supplementary files

MDAR checklist

## Data availability

The current manuscript analyzes publicly available data not generated by the authors.

The following previously published datasets were used:

| Author(s) | Year | Dataset title | Dataset URL | Database and Identifier |
|---|---|---|---|---|
| Siegle et al. | 2021 | Allen Brain Observatory -- Neuropixels Visual Coding [dataset] | https://portal.brain-map.org/circuits-behavior/visual-behavior-neuropixels | Allen Brain Map, visual-behavior-neuropixels |
| Steinmetz N | 2019 | Distributed coding of choice, action and engagement across the mouse brain | http://dx.doi.org/10.6084/m9.figshare.9598406 | figshare, 10.6084/m9.figshare.9598406 |

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
