## [Editor Report · eLife Assessment]

This paper provides a **useful** systematic quantification of the relationship between electrophysiological response properties of single neurons with their position in the brain. The quality of the classification setup is high and the methodology is **solid**.

---

## [Referee Report · Reviewer #1 (Public review)]

Summary:

The paper by Tolossa et al. presents classification studies that aim to predict the anatomical location of a neuron from the statistics of its in-vivo firing pattern. They study two types of statistics (ISI distribution, PSTH) and try to predict the location at different resolutions (region, subregion, cortical layer).

Strengths:

This paper provides a systematic quantification of the single-neuron firing vs location relationship.

The quality of the classification setup seems high.

The paper uncovers that, at the single neuron level, the firing pattern of a neuron carries some information on the neuron's anatomical location, although the predictive accuracy is not high enough to rely on this relationship in most cases.

Weaknesses:

As the authors mention in the Discussion, it is not clear whether the observed differences in firing is epiphenomenal. If the anatomical location information is useful to the neuron, to what extent can this be inferred from the vicinity of the synaptic site, based on the neurotransmitter and neuromodulator identities? Why would the neuron need to dynamically update its prediction of the anatomical location of its pre-synaptic partner based on activity when that location is static, and if that information is genetically encoded in synaptic proteins, etc (e.g., the type of the synaptic site)? Note that the neuron does not need to classify all possible locations to guess the location of its pre-synaptic partner because it may only receive input from a subset of locations. Ultimately, the inability to dissect whether the paper's findings point to a mechanism utilized by neurons or merely represent an epiphenomenon is the main weakness of the curious, though somewhat weak, observations described in this paper.

---

## [Referee Report · Reviewer #2 (Public review)]

Summary:

In this manuscript, Tolossa et al. analyze Inter-spike intervals from various freely available datasets from the Allen Institute and from a dataset from Steinmetz et al.. They show that they can modestly decode between gross brain regions (Visual vs. Hippocampus vs. Thalamus), and modestly separate sub areas within brain regions (DG vs. CA1 or various visual brain areas). The core result is that a multi-layer perceptron trained on the ISI distributions can modestly classify different brain areas and perhaps in a reasonably compelling way generalize across animals. The result is interesting but the exact problem formulation still feels a tad murky to me because I am worried the null is a strawman and I'm unsure if anyone has ever argued for this null hypothesis ("the impact of anatomy on a neuron's activity is either nonexistent or unremarkable"). Given the patterns of inputs to different brain areas and the existence of different developmental origin and different cell types within these areas, I am unclear why this would be a good null hypothesis. Nevertheless, the machine learning is reasonable, and the authors demonstrate that a nonlinear population based classifier can pull out reasonable information about the brain area and layer.

Strengths:

The paper is reasonably well written, and the definitions are quite well done. For example, the authors clearly explained transductive vs. inductive inference in their decoders. E.g., transductive learning allows the decoder to learn features from each animal, whereas inductive inference focuses on withheld animals and prioritizes the learning of generalizable features. The authors walk the reader through various analyses starting as simply as PCA, then finally showing a MLP trained on ISI distributions and PSTHs performs modestly well in decoding brain area. The key is ISI distributions work well in inductive settings for generalizing from one mouse to the other.

Weaknesses:

As articulated in my overall summary, I still found the null hypothesis a tad underwhelming. I am not sure this is really a valid null hypothesis ("the impact of anatomy on a neuron's activity is either nonexistent or unremarkable"), although in the statistical sense it is fine. The authors took on board some of the advice from the first review and clarified the paper but there are portions that are unnecessarily verbose (e.g., "Beyond fundamental scientific insight, our findings may be of benefit in various practical applications, such as the continued development of brain-machine interfaces and neuroprosthetics"). Also, given that ISIs cannot separate between visual areas, why is the statement that these are conserved. I still find it somewhat underwhelming that the thalamus, hippocampus , and visual cortex have different ISI distributions. Multiple researchers have reported similar things in cortex perhaps without the focus on decoding area from these ISI distributions.

All in all, it is an interesting paper with the notion that ISI distributions can modestly predict brain area and layer. It could have some potential for a tool for neuropixels, although this needs to be developed further for this use case.

---

## [Author Response]

The following is the authors’ response to the original reviews

**Reviewer #1 (Recommendations for the authors):**

We appreciate the reviewers' thoughtful comments and suggestions. Below, we provide point-by-point responses to the recommendations and outline the updates made to the manuscript.

(1) Discussion, "the obvious experiment is to manipulate a neuron's anatomical embedding while leaving stimulus information intact." The epiphenomenon can arise from the placement and types of a neuron's neurotransmitters and neuromodulators, too.

The content of vesicles released by a neuron is obviously of great importance in determining postsynaptic impact. However, we’re suggesting that (assuming vesicular content is held constant) the anatomically-relevant patterning of spiking might additionally affect the postsynaptic neuron’s integration of the presynaptic input. To avoid confusion, we updated the text accordingly: “the obvious experiment is to manipulate a neuron's anatomical embedding while minimally impacting external and internal variables, such as stimulus information and levels of neurotransmitters or neuromodulators” (Line 594 - 596).

(2) “In all conditions, the slope of the input duration versus sensitivity line was still positive at 1,800 seconds (Fig. 3B)". This may suggest that the estimate of the calculated statistics (ISI, PSTH) is more reliable with more data, rather than (or in addition to) specific information being extracted from faraway time points. Another potential confound is the training statistics were calculated from all training data, so the test data is a better match to training data when test statistics are calculated from more data. Overall, the validity of the conclusions following this observation is not clear to me.

This is a great point. Accordingly, we revised the text to include this possibility: “Because the training data were of similar duration, this could be explained by either of two possibilities. First, the signal is relatively short, but noisy—in this case, extended sampling will increase reliability. Second, the anatomical signal is, itself, distributed over time scales of tens to hundreds of seconds.” (Line 252 - 255).

(3) "This further suggests that there is a latent neural code for anatomical location embedded within the spike train, a feature that could be practically applied to determining the brain region of a recording electrode without the need for post-hoc histology". The performance of the model at the subregion level, which is a typical level of desired precision in locating cells, does not seem to support such a practical application. Please clarify to avoid confusion.

The current model should not be considered a replacement for traditional methods, such as histology. Our intention is to convey that, with the inclusion of multimodal data and additional samples, a computational approach to anatomical localization has great promise. We updated the manuscript to clarify this point: “While significantly above chance, the structure-level model still lacks the accuracy for immediate practical application. However, it is highly likely that the incorporation of datasets with diverse multi-modal features and alternative regions from other research groups will increase the accuracy of such a model. In addition, a computational approach can be combined with other methods of anatomical reconstruction.” (Line 355 - 359).

Additionally, we directly addressed this point in our original manuscript (Discussion section: Line 498 - 505 in the current version). Furthermore, following the release of our preprint, independent efforts have adopted a multimodal strategy with qualitatively similar results (Yu et al., 2024). Other recent work expands on the idea of utilizing single-neuron features for brain region/structure characterization (La Merre et al., 2024).

Yu, H., Lyu, H., Xu, E. Y., Windolf, C., Lee, E. K., Yang, F., ... & Hurwitz, C. (2024). In vivo cell-type and brain region classification via multimodal contrastive learning. bioRxiv, 2024-11.

Le Merre, P., Heining, K., Slashcheva, M., Jung, F., Moysiadou, E., Guyon, N., ... & Carlén, M. (2024). A Prefrontal Cortex Map based on Single Neuron Activity. bioRxiv, 2024-11.

(4) "These results support the notion the meaningful computational division in murine visuocortical regions is at the level of VISp versus secondary areas.". The use of the word "meaningful" is vague and this conclusion is not well justified because it is possible that subregions serve different functional roles without having different spiking statistics.

Precisely! It is well established that different subregions serve different functional purposes - but they do not necessitate different regional embeddings. It is important to note the difference between stimulus encoding and the embedding that we are describing. As a rough analogy, the regional embedding might be considered a language, while the stimulus is the content of the spoken words. However, to avoid vague words, we revised the sentence to “These results suggest that the computational differentiability of murine visuocortical regions is at the level of VISp versus secondary areas.” (Line 380 - 381)

(5) Figure 3D left/right halves look similar. A measure of the effect size needs to accompany these p-values.

We assume the reviewer is referring to Figure 3E. Although some of the violin plots in Figure 3E look similar, they are not identical. In the revision, we include effect sizes in the caption.

(6) Figure 3A, 3F: Could uncertainty estimates be provided?

Yes. We added uncertainty estimates to the text (Line 272 - 294) and to the caption of Figure S2, which displays confusion matrices corresponding to Figure 3A. The inclusion of similar estimates for 3F would be so unwieldy as to be a disservice to the reader—there are 240 unique combinations of stimulus parameters and structures. In the context of the larger figure, 3F serves to illustrate a relationship between stimulus, region, and the anatomical embedding.

(7) Page 21. "semi-orthogonal". Please reword or explain if this usage is technical.

We replaced “semi-orthogonal” with “dissociable” (Line 549).

(8) Page 11, "This approach tested whether..." Unclear sentence. Please reword.

We changed “This approach tested whether the MLP’s performance depended on viewing the entire ISI distribution or was enriched in a subset of patterns” to “This approach identified regions of the ISI distribution informative for classification” (Line 261).

**Reviewer #2 (Recommendations for the authors):**

We appreciate the reviewer’s comments and summary of the results. We agree that the introductory results (Figs. 1-3) are not particularly compelling when considered in isolation. They provide a baseline of comparison for the subsequent results. Our intention was to approach the problem systematically, progressing from well-established, basic methods to more advanced approaches. This allows us to clearly test a baseline and avoid analytical leaps or untested assumptions. Specifically:

● Figure 1 provides an evaluation of the standard dimensionality reduction methods. As expected, these methods yield minimal results, serving as a clear baseline. This is consistent, for example, with an understanding of single units as rate-varying Poisson processes.

● Figures 2 and 3 then build upon these results with spiking features frequent in neuroscience literature such as firing rate, coefficient of variation, etc using linear supervised and more detailed spiking features such as ISI distribution using nonlinear supervised machine learning methods.

By starting from the standpoint of the status quo, we are better able to contextualize the significance of our later findings in Figures 4–6.

Response to Specific Points in the Summary

(6) Separability of VISp vs. Secondary Visual AreasI found the entire argument about visual areas somewhat messy and unclear. The stimuli used might not drive the secondary visual areas particularly well and might necessitate task engagement.

We appreciate your feedback that the dissection of visual cortical structures is unclear. To summarize, as shown in the bottom three rows of Figure 6, there is a notable lack of diagonality in visuocortical structures. This means that our model was unable to learn signatures to reliably predict these classes. In contrast, visuocortical layer is returned well above chance, and superstructures (primary and secondary areas) are moderately well identified, albeit still well above chance.

Consider a thought experiment, if Charlie Gross had not shown faces to monkeys to find IT, or Newsome and others shown motion to find MT and Zeki and others color stimuli to find V4, we would conclude that there are no differences.

The thought experiment is misleading. The results specifically do not arise from stimulus selectivity—much of Newsome’s own work suggests that the selectivity of neurons in IT etc. is explained by little more than rate varying Poisson processes. In this case, there should be no fundamental anatomical difference in the “language” of the neurons in V4 and IT, only a difference in the inputs driving those neurons. In contrast, our work suggests that the “language” of neurons varies as a function of some anatomical divisions. In other words, in contrast to a Poisson rate code, our results predict that single neuron spike patterns might be remarkably different in MT and IT— and that this is not a function of stimulus selectivity. Notably, the anatomical (and functional) division between V1 and secondary visual areas does not appear to manifest in a different “language”, thus constituting an interesting result in and of itself.

We regret a failure to communicate this in a tight and compelling fashion on the first submission, but hope that the revision is limpid and accessible.

Barberini, C. L., Horwitz, G. D., & Newsome, W. T. (2001). A comparison of spiking statistics in motion sensing neurones of flies and monkeys. Motion Vision: Computational, Neural, and Ecological Constraints, 307-320.

Bair, W., Zohary, E., & Newsome, W. T. (2001). Correlated firing in macaque visual area MT: time scales and relationship to behavior. Journal of Neuroscience, 21(5), 1676-1697.

Similarly, why would drifting gratings be a good example of a stimulus for the hippocampus, an area thought to be involved in memory/place fields?

The results suggest that anatomical “language” is not tied to stimuli. It is imperative to recall that neurons are highly active absent experimentally imposed stimuli, such as when an animal is at rest, when an animal is asleep, and when an animal is in the dark (relevant to visual cortices). With this in mind, also recall that, despite the lack of stimuli tailored to the hippocampus, neurons therein were still reliably separable from neurons in seven nuclei in the thalamus, 6 of which are not classically considered visual regions. Should these regions (including hippocampus) have been inert during the presentation of visual stimuli, there would have been very little separability.

(7) Generalization across laboratories“[C]omparison across laboratories was somewhat underwhelming. It does okay but none of the results are particularly compelling in terms of performance.

Any result above chance is a rejection of the null hypothesis: that a model trained on a set of animals in Laboratory A will be ineffective in identifying brain regions when tested on recordings collected in Laboratory B (in different animals and under different experimental conditions). As an existence proof, the results suggest conserved principles (however modest) that constrain neuronal activity as a function of anatomy. That models fail to achieve high accuracy (in this context) is not surprising (given the limitations of available recordings)---that models achieve anything above chance, however, is.

Thus, after reading the paper many times, I think part of the problem is that the study is not cohesive, and the authors need to either come up with a tool or demonstrate a scientific finding.

We demonstrate that neuronal spike trains carry robust anatomical information. We developed an ML architecture for this and that architecture is publicly available.

They try to split the middle and I am left somewhat perplexed about what exact scientific problem they or other researchers are solving.

We humbly suggest that the question of a neurons “language” is highly important and central to an understanding of how brains work. From a computational perspective, there is no reason for a vast diversity of cell types, nor a differentiation of the rules that dictate neuronal activity in one region versus another. A Turing Complete system can be trivially constructed from a small number of simple components, such as an excitatory and inhibitory cell type. This is the basis of many machine learning tools.

Please do not confuse stimulus specificity with the concept of a neuron’s language. Neurons in VISp might fire more in response to light, while those in auditory cortex respond to sound. This does not mean that these neurons are different - only that their inputs are. Given the lack of a literature describing our main effect—that single neuron spiking carries information about anatomical location—it is difficult to conclude that our results are either commonplace or to be expected.

I am also unsure why the authors think some of these results are particularly important.

See above.

For instance, has anyone ever argued that brain areas do not have different spike patterns?

Yes. In effect, by two avenues. The first is a lack of any argument otherwise (please do not conflate spike patterns with stimulus tuning), and the second is the preponderance of, e.g., rate codes across many functionally distinct regions and circuits.

Is that not the premise for all systems neuroscience?

No. The premise for all systems neuroscience (from our perspective) is that the brain is (a) a collection of interacting neurons and (b) the collective system of neurons gives rise to behavior, cognition, sensation, and perception. As stated above, these axiomatic first principles fundamentally do not require that neurons, as individual entities, obey different rules in different parts of the brain.

I could see how one could argue no one has said ISIs matter but the premise that the areas are different is a fundamental part of neuroscience.

Based on logic and the literature, we fundamentally disagree. Consider: while systems neuroscience operates on the principle that brain regions have specialized functions, there is no a priori reason to assume that these functions must be reflected in different underlying computational rules. The simplest explanation is that a single language of spiking exists across regions, with functional differences arising from processing distinct inputs rather than fundamentally different spiking rules. For example, an identical spike train in the amygdala and Layer 5 of M1 would have profoundly different functional impacts, yet the spike timing itself could be identical (even as stimulus response). Until now, evidence for region-specific spiking patterns has been lacking, and our work attempts to begin addressing this gap. There is extensive further work to be conducted in this space, and it is certain that models will improve, rules will be clarified, and mechanisms will be identified.

Detailed major comments(1) Exploratory trends in spiking by region and structure across the population:The argument in this section is that unsupervised analyses might reveal subtle trends in the organization of spiking patterns by area. The authors show 4 plots from t-SNE and claim to see subtle organization. I have concerns. For Figure 1C, it is nearly impossible to see if a significant structure exists that differentiates regions and structures. So this leads certain readers to conclude that the authors are looking at the artifactual structure (see Chari et al. 2024) - likely to contribute to large Twitter battles. Contributing to this issue is that the hyperparameter for tSNE was incorrectly chosen. I do think that a different perplexity should be used for the visualization in order to better show the underlying structure; the current visualization just looks like a single "blob". The UMAP visualizations in the supplement make this point more clearly. I also think the authors should include a better plot with appropriate perplexity or not include this at all. The color map of subtle shades of green and yellow is hard to see as well in both Figure S1 and Figure 1.

In response to the feedback, we replaced t-SNE/UMAP with LDA, while keeping PCA for dimensionality reduction.

As stated in the original methods, t-SNE/UMAP hyperparameters were chosen based on the combination that led to the greatest classifiable separability of the regions/structures in the space (across a broad range of possible combinations). It just so happens that the maximally separable structure from a regions/structures perspective is the “blob”. This suggests that perhaps the predominant structure the t-SNE finds in the data is not driven by anatomy. If we selected hyperparameters in some other way that was not based specifically on regions/structures (e.g. simple visual inspection of the plots) the conformation would of course be different and not blob-like. However, we removed the t-SNE and UMAP to avoid further confusion.

The “muddy appearance” is not an issue with the color map. As seen in Figure 1B, the chosen colors are visibly distinct. Figure 1C (previous version) appeared muddy yellow/green because of points that overlap with transparency, resulting in a mix of clearly defined classes (e.g., a yellow point on top of a blue point creating green). This overlap is a meaningful representation of the separability observed in this analysis. We also tried using 2D KDE for visualization, but it did not improve the impression of visual separability.

We are removing p-values from the figures because they lead to the impression that we over-interpret these results quantitatively. However, we calculated p-values based on label permutation similar to the way R2 suggests (see previous methods). The conflation with the Wasserstein distances is an understandable misunderstanding. These are unrelated to p-values and used for the heatmaps in S1 only (see previous methods).

Instead of p-values, we now use the adjusted rand index, which measures how accurately neurons within the same region are clustered together (see Line 670 - 671, Figure 1C, and Figure S1) (Hubert & Arabie 1985). This quantifies the extent to which the distribution of points in dimensionally-reduced space is shaped by region/structure.

Hubert, L., & Arabie, P. (1985). Comparing partitions. Journal of Classification, 2(1), 193–218. https://doi.org/10.1007/BF01908075

(2) Logistic classifiers:The results in this section are somewhat underwhelming. Accuracy is around 40% and yes above chance but I would be very surprised if someone is worried about separating visual structures from the thalamus. Such coarse brain targeting is not difficult. If the authors want to include this data, I recommend they show it as a control in the ISI distribution section. The entire argument here is that perhaps one should not use derived metrics and a nonlinear classifier on more data is better, which is essentially the thrust of the next section.

As outlined above, our work systematically increases in model complexity. The logistic result is an intermediate model, and it returns intermediate results. This is an important stepping stone between the lack of a result based on unsupervised linear dimensionality reduction and the performance of supervised nonlinear models.

From a purely utilitarian perspective, the argument could be framed as “one should not use derived metrics, and a nonlinear classifier on more data is better.” However, please see all of our notes above.

(3) MLP classifiers:Even in this section, I was left somewhat underwhelmed that a nonlinear classifier with large amounts of data outperforms a linear classifier with small amounts of data. I found the analysis of the ISIs and which timescales are driving the classifier interesting but I think the classifier with smoothing is more interesting. So with a modest chance level decodability of different brain areas in the visual system, I found it somewhat grandiose to claim a "conserved" code for anatomy in the brain. If there is conservation, it seems to be at the level of the coarse brain organization, which in my opinion is not particularly compelling.

The sample size used for both the linear and nonlinear classifiers is the same; however, the nonlinear classifier leverages the detailed spiking time information from ISIs. Our goal here was to systematically evaluate how classical spike metrics compare to more detailed temporal features in their ability to decode brain areas. We chose a linear classifier for spike metrics because, with fewer features, nonlinear methods like neural networks often offer very modest advantages over linear methods, less interpretability, and are prone to overfitting.

Respectfully, we stand by our word choice. The term “conserved” is appropriate given that our results hold appreciably, i.e., statistically above chance, across animals.

(4) Generalization section:The authors suggest that a classifier learned from one set of data could be used for new data. I was unsure if this was a scientific point or the fact that they could use it as a tool.

It can be both. We are more driven by the scientific implications of a rejection of the null.

Is the scientific argument that ISIs are similar across areas even in different tasks?

It appears so - despite heterogeneity in the tuning of single neurons, their presynaptic inputs, and stimuli, there is identifiable information about anatomical location in the spike train.

Why would one not learn a classifier from every piece of available data: like LFP bands, ISI distributions, and average firing rates, and use that to predict the brain area as a comparison?

Because this would obfuscate the ability to conclude that spike trains embed information about anatomy.

Considering all features simultaneously and adding additional data modalities—such as LFP bands and spike waveforms—has potential to improve classification accuracy at the cost of understanding the contribution of each feature. The spike train as a time series is the most fundamental component of neuronal communication. As a result, this is the only feature of neuronal activity of concern for the present investigation.

Or is the argument that the ISIs are a conserved code for anatomy? Unfortunately, even in this section, the data are underwhelming.

We appreciate the reviewer’s comments, but arrive at a very different conclusion. We were quite surprised to find any generalizability whatsoever.

Moreover, for use as a tool, I think the authors need to seriously consider a control that is either waveforms from different brain areas or the local field potentials. Without that, I am struggling to understand how good this tool is. The authors said "because information transmission in the brain arises primarily from the timing of spiking and not waveforms (etc)., our studies involve only the timestamps of individual spikes from well-isolated units ". However, we are not talking about information transmission and actually trying to identify and assess brain areas from electrophysiological data.

While we are not blind to the “tool” potential that is suggested by our work, this is not the primary motivation or content in any section of the paper. As stated clearly in the abstract, our motivation is to ask “whether individual neurons [...] embed information about their own anatomical location within their spike patterns”. We go on to say “This discovery provides new insights into the relationship between brain structure and function, with broad implications for neurodevelopment, multimodal integration, and the interpretation of large-scale neuronal recordings. Immediately, it has potential as a strategy for in-vivo electrode localization.” Crucially, the last point we make is a nod to application. Indeed, our results suggest that in-vivo electrode localization protocols may benefit from the incorporation of such a model.

In light of the reviewer’s concerns, we have further dampened the weight of statements about our model as a consumer-ready tool.

Example 1: The final sentence of the abstract now reads: “Computational approximations of anatomy have potential to support in-vivo electrode localization.”

Example 2: The results sections now contains the following text: “While significantly above chance, the structure-level model still lacks the accuracy for immediate practical application. However, it is highly likely that the incorporation of datasets with diverse multi-modal features and alternative regions from other research groups will increase the accuracy of such a model. In addition, a computational approach can be combined with other methods of anatomical reconstruction.” (Line 355 - 359).

Example 3: We replaced the phrase "because information transmission in the brain arises primarily from the timing of spiking and not waveforms (etc) " with the phrase “because information is primarily encoded by the firing rate or the timing of spiking and not waveforms (etc)” (Line 116 - 118).

(5) Discussion section:In the discussion, beginning with "It is reasonable to consider . . ." all the way to the penultimate paragraph, I found the argumentation here extremely hard to follow. Furthermore, the parts of the discussion here I did feel I understood, I heavily disagreed with. They state that "recordings are random in their local sampling" which is almost certainly untrue when it comes to electrophysiology which tends to oversample task-modulated excitatory neurons (https://elifesciences.org/articles/69068). I also disagree that "each neuron's connectivity is unique, and vertebrate brains lack 'identified neurons' characteristic of simple organisms. While brains are only eutelic and "nameable" in only the simplest organisms (*C. elegans*), cell types are exceedingly stereotyped in their connectivity even in mammals and such connectivity defines their computational properties. Thus I don't find the premise the authors state in the next sentence to be undermined ("it seems unlikely that a single neuron's happenstance imprinting of its unique connectivity should generalize across stimuli and animals"). Overall, I found this subsection to rely on false premises and in my opinion it should be removed.

At the suggestion of R2, we removed the paragraph in question. However, we would like to address some points of disagreement:

We agree that electrophysiology, along with spike-sorting, quality metrics, and filtering of low-firing neurons, leads to oversampling of task-modulated neurons. However, when we stated that recordings are random in their local sampling, we were referring to structural (anatomical) randomness, not functional randomness. In other words, the recorded neurons were not specifically targeted (see below).

Electrode arrays, such as Neuropixels, record from hundreds of neurons within a small volume relative to the total number of neurons and the volume of a given brain region. For instance, the paper R2 referenced includes a statement supporting this: “... assuming a 50-μm ‘listening radius’ for the probes (radius of half-cylinder around the probe where the neurons’ spike amplitude is sufficiently above noise to trigger detection) …, the average yield of 116 regular-spiking units/probe (prior to QC filtering) would imply a density of 42,000 neurons/mm³, much lower than the known density of ~90,000 neurons/mm³ for excitatory cells in mouse visual cortex….”

If we take the estimated volume of V1 to be approximately 3 mm³, this region could theoretically be subdivided into multiple cylinders with a 100-μm diameter. While stereotaxic implantation of the probe mitigates some variability, the natural anatomical variability across individual animals introduces spatially random sampling. This was the randomness we were referring to, and thus, we disagree with the assertion that our claim is “almost certainly untrue.”

Additionally, each cortical pyramidal neuron is understood to have ~ 10,000 presynaptic partners. It is highly unlikely that these connections are entirely pre-specified, perfectly replicated within the same animal, and identical across all members of species. Further, there is enormous diversity in the activity properties of even neighboring cells of the same type. Consider pyramidal neurons in V1. Single neuron firing rates are log normally distributed, there are many of combinations of tuning properties (i.e., direction, orientation) that must occupy each point in retinotopic space, and there is powerful experience dependent change in the connectivity of these cells. We suggest that it is inconceivable that any two neurons, even within a small region of V1, have identical connectivity.

Minor Comments:(1) Although the description of confusion matrices is good from a didactic perspective, some of this could be moved to methods to simplify the paper.

We thank the reviewer for the suggestion. However, given the broad readership of eLife, we gently suggest that confusion matrices are not a trivial and universally appreciated plotting format. For the purpose of accessibility, a brief and didactic 2-sentence description will make the paper far more comprehensible to many readers at little cost to experts.

(2) Figure 3A: It is concluded in their subsequent figure that the longer the measured amount of time, the better the decoding performance. Thus it makes sense why the average PSTHs do not show significant decoding of areas or structures

That is a good observation. However, all features were calculated from the same duration of data, except in Figure 3B, where we tested the effect of duration. The averaged PSTH was calculated from the same length of data as the ISI distribution and binned to have the same number of feature lengths as the ISI distribution (refer to Methods section). Therefore, we interpreted this as an indication of information degradation through averaging, rather than an effect of data length (Line 234 - 237).

(3) Figure 3D: A Gaussian is used to fit the ISI distributions here but ISI distributions do not follow a normal distribution, they follow an inverse gamma distribution.

We agree with the reviewer and we are familiar with the literature that the ISI distribution is best fitted by a gamma family distribution (as a recent, but not earliest example: Li et al. 2018). However, we did not fit a gaussian (or any distribution) to the data, we just calculated the sample mean and variance. Reporting sample mean and variance (or standard deviation) is not something that is only done for Gaussian distributions. They are broadly used metrics that simply have additional intrinsic meaning for Gaussian distributions. We used the schematic illustration in Fig 3D because mean and variance are much more familiar in Gaussian distribution context, but ultimately that does not affect our analyses in Fig 3 E-F. Alternatively, the alpha and beta intrinsic parameters of a gamma distribution could have been used, but they are known by a much smaller portion of neuroscientists.

Li, M., Xie, K., Kuang, H., Liu, J., Wang, D., Fox, G. E., ... & Tsien, J. Z. (2018). Spike-timing pattern operates as gamma-distribution across cell types, regions and animal species and is essential for naturally-occurring cognitive states. Biorxiv, 145813(10.1101), 145813.

(4) Figure 3G: Something is wrong with this figure as each vertical bar is supposed to represent a drifting grating onset but yet, they are all at 5 hz despite the PSTH being purportedly shown at many different frequencies from 1 to 15 hz.

We appreciate your attention to detail, but we are not representing the onset of individual drifting gratings in this. We just meant to represent the overall start\end of the drifting grating session. We did not intend to signal the temporal frequency of the drifting gratings (or the spatial frequency, orientation, or contrast).